# Voronoi Tessellation-based Confidence Decision Boundary Visualization to Enhance Understanding of Active Learning

## Abstract

The current visualizations used in active learning fail to capture the cumulative effect of the model in the active learning process, making it difficult for researchers to effectively observe and analyze the practical performance of different query strategies. To address this issue, we introduce the *confidence decision boundary visualization*, which is generated through Voronoi tessellation and evaluated using ridge confidence. This allows better understanding of selection strategies used in active learning. This approach enhances the information content in boundary regions where data distribution is sparse. Based on the confidence decision boundary, we created a series of visualizations to evaluate active learning query strategies. These visualizations capture nuanced variations regarding how different selection strategies perform sampling, the characteristics of points selected by various methods, and the impact of newly sampled points on the model. This enables a much deeper understanding of the underlying mechanisms of existing query strategies.

## 1 Introduction

Active learning (AL) (Settles, 2009) is semi-supervised machine learning approach that aims to minimize labeling costs by identifying the most informative samples in a set of unlabelled data for labeling, thereby improving learning efficiency with a limited amount of labeled data. AL techniques have been shown to be effective in various domains, such as image classification (Joshi et al., 2009), medical diagnosis (Budd et al., 2021), and natural language processing (Dor et al., 2020). Despite their effectiveness, understanding and analyzing the rationale behind the data sampled by different strategies remains a significant challenge.

Existing approaches to AL visualization primarily focus on illustrating the selection process and the spatial distribution of data points in a 2D space. Mac Namee et al. (2010) employed scatter plots to visualize the relationships between selected points and the remaining pool, highlighting their uncertainty or diversity levels. Building on this, Liao et al. (2016) introduced iso-contours to enhance the scatter plot representation. Huang et al. (2017) developed an interactive visualization tool for text classification, using 2D scatter plots to depict labeled and unlabeled points, facilitating point selection for labeling. Hilasaca et al. (2021) proposed a method combining label propagation and clustering-based sample selection, where multidimensional projections were utilized to map data similarity into 2D space, providing a more comprehensive understanding of the labeling stage. Other approaches focus on visualizing sampled data distributions, individual samples, and basic performance metrics (Agarwal et al., 2020; Pinsler et al., 2019; Liu et al., 2021), or on illustrating the impact of training samples on decision boundaries in simple models and specific AL query scenarios (Joshi et al., 2009; Tharwat & Schenck, 2023). However, these methods primarily capture relationships within the current round of sampling or between the model and data at a single stage. They fail to account for the iterative accumulation of the model's training data and its influence in AL, thus providing limited insight into observing the dynamics of query strategies.

The decision boundary provides an intuitive and rich representation of how a model separates different classes within the data (Migut et al., 2015). As a result, uncertainty-based methods, a major category of query strategies in AL, focus on selecting points near the decision boundary (Settles,

2009). However, visualizing the decision boundary in AL is much more challenging than a simple performance evaluation. These difficulties arise mainly from approximating high-dimensional decision boundaries within low-dimensional spaces. Most prior work on decision boundary visualization has primarily focused on classification tasks (Rodrigues et al., 2018; Somepalli et al., 2022; Migut et al., 2015; 2011; Melnik, 2002). For example, Somepalli et al. (2022) used the difference between two sets of data features as coordinate dimensions to visualize decision regions, which is not well-suited for the large pool datasets commonly encountered in AL scenarios. Rodrigues et al. (2018) projected points from the original data space onto a two-dimensional grid composed of pixel blocks, similar to an image. However, this approach can distort the representation of distances between data points, making it unsuitable for analyzing query strategies that rely on distance metrics. Migut et al. (2015) first introduced the use of Voronoi tessellation to partition the reduced-dimensional space and generate decision boundaries. However, their method treated multi-class tasks as collections of binary classifications, significantly limiting its applicability in AL visualization.

A common issue with the current decision boundary visualization methods is their inability to capture the varying levels of complexity and uncertainty across different regions of the boundary. These methods treat all sections of the decision boundary uniformly, despite the fact that under different data distributions, the sections of the boundary that approach the true boundary can vary. To address these challenges, we propose a novel **confidence decision boundary visualization** method based on Voronoi tessellation (Aurenhammer, 1991) for AL. Voronoi tessellation is widely used to partition boundaries between different classes (Migut et al., 2015) or clusters (Chen et al., 2021). Our method leverages Voronoi tessellation to assign each labeled data point learned by the model to a cell, which can be regarded as a set of similar points (De Berg, 2000), with the labeled point serving as a representative of this set based on nearest-neighbor relationships. This approach not only visualizes the model's understanding of unlabeled data in the pool dataset but also avoids data overlap and eliminates undefined blank regions often caused by sparsity near the decision boundary. We recognize that the low-dimensional representation of the decision boundary depends on the chosen dimensionality reduction method, and different sections of the decision boundary contain varying levels of information based on data distribution. To capture these variations in information, we decompose the decision boundary into multiple predicted ridges, each evaluated using a ridge confidence metric, which can quickly identify the regions of the decision boundary that are closest to the true boundary.

Using this more granular partitioning of the decision boundary, we conducted a series of visualization experiments and analysis in two datasets MNIST (LeCun, 1998) and CIFAR-10 (Krizhevsky et al., 2009), specifically to observe different pool-based AL query strategies, addressing the lack of visual analysis in AL. Our key findings are as follows:

- Our visualization uncovers the distinct sampling behaviors of entropy-based methods, highlighting the impact of incorporating Monte Carlo dropout and Bayesian Convolutional Neural Networks. Furthermore, it preliminarily identified two trends in uncertainty - uncertainty from insufficient training samples can be reduced by concentrated sampling, while uncertainty in noisy regions is harder to resolve due to mixed features from multiple classes.

- Through visualization, we observed that least confidence sampling and margin sampling select high uncertainty data, which consist of a mixture of various uncertainty types mentioned above. As the model's performance improves, the proportion of noise data points with high uncertainty tends to increase.

- Our visualization compared three different diversity methods and revealed that KCenter-Greedy is influenced by the model's bias in understanding the data distribution, leading to imbalanced sampling across classes.

## 2 CONFIDENCE DECISION BOUNDARY

Our approach utilizes Voronoi tessellation (Aurenhammer, 1991) to divide the 2D data space and assign confidence values to different segments of the decision boundary, highlighting the variations in boundary certainty across regions.

## 2.1 Voronoi Tessellation

A Voronoi tessellation is a geometric structure that partitions a space based on the proximity of points (Aurenhammer, 1991). Given a set of points $\mathcal{P} = \{\boldsymbol{p}_1, \ldots, \boldsymbol{p}_n\}$ in $\mathbb{R}^d$, each point $\boldsymbol{p}_i$ has an associated Voronoi cell $V(\boldsymbol{p}_i)$, which contains all points $\boldsymbol{x} \in \mathbb{R}^d$ that are closer to $\boldsymbol{p}_i$ than any other point $\boldsymbol{p}_j \in \mathcal{P}, j \neq i$. Formally:

$$V(\boldsymbol{p}_i) = \left\{ \boldsymbol{x} \in \mathbb{R}^d \mid \|\boldsymbol{x} - \boldsymbol{p}_i\| \leq \|\boldsymbol{x} - \boldsymbol{p}_j\|, \forall \boldsymbol{p}_j \neq \boldsymbol{p}_i \right\} \tag{1}$$

A Voronoi tessellation provides insight into the local influence of each point in $\mathcal{P}$ by partitioning the feature space based on nearest-neighbor relationships (Aurenhammer, 1991). In a lower-dimensional projection, these cells reflect the structure of the original feature space while preserving the local neighborhood relationships. The Voronoi ridges that divide these cells in the lower-dimensional projection also carry rich information from the original high-dimensional data. Together, they represent the relationships between data points and the underlying spatial structure.

## 2.2 Confidence Decision Boundary using Voronoi Tessellation

We use features extracted from the original space after dimensionality reduction to construct a 2D Voronoi diagram, where each point in the Voronoi tessellation represents a data instance. Based on the predicted labels of all points $\hat{\mathcal{Y}} = \{\hat{y}_i \mid \boldsymbol{p}_i \in \mathcal{P}\}$, each Voronoi ridge between two points $\boldsymbol{p}_i$ and $\boldsymbol{p}_j$, if $\hat{y}_i \neq \hat{y}_j$, the ridge is identified as a predicted ridge. The set of all predicted ridges forms the decision boundary $\mathcal{B}_{\text{pred}} = \{\text{ridge}(\boldsymbol{p}_i, \boldsymbol{p}_j) \mid \hat{y}_i \neq \hat{y}_j, \boldsymbol{p}_i, \boldsymbol{p}_j \in \mathcal{P}\}$ of the model in the current 2D feature space for the given data distribution.

Since points on either side of the predicted ridges have different probabilities of belonging to each class, different sections of the decision boundary carry varying degrees of informative value based on differences in prediction confidence. Thus, treating all sections of the decision boundary as equally informative can cause observers to miss critical insights.

To address this, we propose the concept of predicted ridge confidence. This confidence reflects the uncertainty in predictions along the ridge, indicating how distinct the predictions are on either side and the reliability of the predicted ridges. For a ridge between two points $\boldsymbol{p}_i$ and $\boldsymbol{p}_j$, this ridge is considered part of the decision boundary, with the surrounding points more sparsely distributed compared to other regions. To estimate the confidence of this ridge based on the points it separates, we leverage the property of Voronoi cells, where all points in $V(\boldsymbol{p}_i)$ can be represented by point $\boldsymbol{p}_i$. Using the model's understanding of the points $\boldsymbol{p}_i$ and $\boldsymbol{p}_j$, the predicted ridge confidence $C_{\text{pred}}$ is defined as:

$$C_{\text{pred}} = 1 - \sum_{k=1}^{K} P(\hat{y}_i = k) P(\hat{y}_j = k) \tag{2}$$

where $K$ is the number of classes, and $P(\hat{y}_i = k)$ and $P(\hat{y}_j = k)$ are the predicted probabilities for class $k$ at points $\boldsymbol{p}_i$ and $\boldsymbol{p}_j$, respectively. The algorithm for generating the confidence decision boundary can be found in Appendix Algorithm 1.

To assist observers in better understanding the decision boundary and how it evolves across different models and datasets, we additionally recognize the ground truth boundary. Using the true labels $y_i$ for each point $\boldsymbol{p}_i$, the Voronoi ridges where neighboring cells have different true labels are defined as ground truth ridges. The collection of all such ridges forms the ground truth boundary:

$$\mathcal{B}_{\text{gt}} = \{\text{ridge}(\boldsymbol{p}_i, \boldsymbol{p}_j) \mid y_i \neq y_j, \boldsymbol{p}_i, \boldsymbol{p}_j \in \mathcal{P}\} \tag{3}$$

## 3 Visualization Setup

### 3.1 Datasets and Models

**MNIST** (LeCun, 1998) We used a classical Convolutional Neural Network (CNN) (Chollet, 2015) model to implement AL strategies. The dataset was split into 50,000 images for the AL pool, with 10,000 images each for validation and testing. AL began with 10 labeled samples to initialize the model, followed by querying 20 samples per round over 30 iterations.

**CIFAR-10** (Krizhevsky et al., 2009) We employed the Vision Transformer (ViT) model, using 16x16 input patches with a base architecture pre-trained on ImageNet-21k (Wightman, 2019). The dataset was divided into 40,000 images for the AL pool and 10,000 each for validation and testing. The model was initialized with 10 labeled samples, then querying 40 samples per round over 12 iterations in the AL process.

## 3.2 ACTIVE LEARNING QUERY STRATEGIES

We evaluated eight widely-used AL strategies, broadly categorized into uncertainty-based and diversity-based methods. The uncertainty-based methods include **Entropy Sampling** (Joshi et al., 2009), which selects instances with the highest uncertainty measured by information entropy; **Least Confidence** Lewis (1995), querying samples with the lowest prediction confidence; **Margin Sampling** Campbell et al. (2000), focusing on instances with the smallest margin between the two most likely classes; **Entropy Sampling Dropout** Ren et al. (2021), combining Monte Carlo Dropout with entropy-based sampling to account for both model and predictive uncertainty; and **BALD Dropout** Houlsby et al. (2011), using Bayesian Active Learning by Disagreement with Dropout to maximize information gain. The diversity-based methods include **Random Sampling**, a baseline method selecting samples randomly to ensure diverse representation without model bias; **KMeans Sampling** Nguyen & Smeulders (2004), selecting samples from diverse clusters in feature space; and **K-Center Greedy** Sener & Savarese (2018), maximizing feature space coverage by choosing samples furthest from the labeled set.

Our experimental results on both MNIST and CIFAR-10 are presented in Figure 1a and Figure 1b, respectively. On MNIST, KMeans performs significantly worse than random sampling, while on CIFAR-10, it slightly outperforms random. In contrast, KCenterGreedy shows the opposite trend, performing better on MNIST but worse on CIFAR-10. This inconsistency may be due to both methods relying on distance calculations in the feature space, which can be distorted by the model's biased understanding of the data distribution with the limited sampling budget. Entropy and Entropy Dropout exhibit similar trends, with nearly identical accuracy per round. Both methods, along with KCenterGreedy, experience a sharp accuracy drop in the first round on CIFAR-10, likely due to selecting difficult samples early on, which hinders the model's ability to quickly build an understanding of the data. Margin and Least Confidence sampling consistently outperform other methods on both datasets, achieving greater overall accuracy improvements and reaching the highest accuracy in training. Based on the experimental results and the characteristics of each strategy, we categorized the eight strategies into three groups for separate discussion.

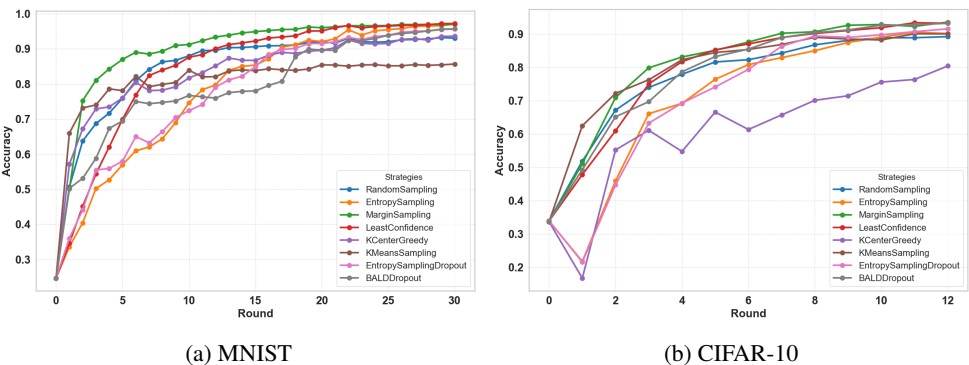

(a) MNIST                    (b) CIFAR-10

Figure 1: The performance of models for different strategies over rounds

## 3.3 VISUALIZATION MODEL

During the AL process, we visualized the dimensionality-reduced features extracted by the model from the pool dataset of MNIST at different iterations using Voronoi tessellation as shown in Figure 2. The black dashed lines in the figure outline the true boundary obtained from the ground truth labels. As the model's performance improves along with the increase of the iteration, the features extracted by the model become more distinct in the two-dimensional space, leading to clearer and

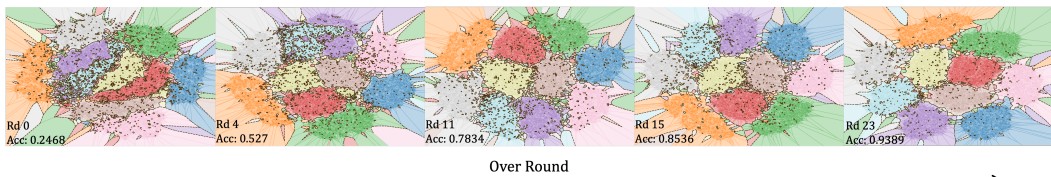

Figure 2: 2D feature extracted from different classification models by using entropy sampling on MNIST during the AL process

more streamlined ground truth boundaries. To extract features that not only preserve the key characteristics of the original data but also exhibit maximal separability between different classes in the two-dimensional space, we subsequently trained an individual visualization model with the same architecture as the AL model on the entire pool dataset to achieve optimal accuracy. This approach provides a fixed 2D feature distribution, facilitating a consistent comparison and analysis of various query strategies through visual examination. Regarding the potential spatial distortion caused by dimensionality reduction techniques on the Voronoi diagram and decision boundary, we conducted two quantitative evaluations: Correlation Test (Smyth et al., 2000; Namee & Delany, 2010) and Local Structure Preservation (Huang et al., 2022), on the t-SNE method used in this study. The results, detailed in the Appendix (Table 1 and Figure 9), demonstrate that the impact of such distortion is limited, indicating that its effects on the Voronoi diagram and decision boundary are relatively small.

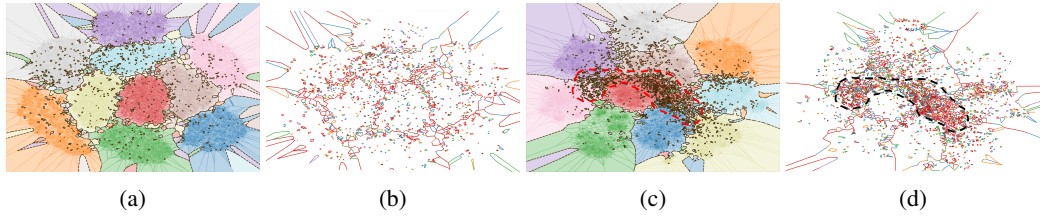

(a)                    (b)                    (c)                    (d)

Figure 3: (a) Ground truth boundary for MNIST, derived from the 2D features extracted by the visualisation model with an accuracy of 0.9907. (b) The final round Confidence Decision Boundary for MNIST, generated by the Entropy-based model. (c) Ground truth boundary for CIFAR-10, derived from the 2D features extracted by the visualisation model with an accuracy of 0.9831. (d) The final round Confidence Decision Boundary for CIFAR-10, generated by the Margin-based model. In (a) and (c), the brown dashed line represents the ground truth boundary separating different classes.

Figures 3a and 3c show the ground truth boundaries based on the 2D features from the visualization models, whereas Figures 3b and 3d visualize the decision boundaries with different confidence interval from the final AL round for MNIST and CIFAR-10, respectively. It is evident that predicted ridges with higher confidence align more closely with the ground truth boundaries as presented in Figures 3b and 3d, where red lines represent high confidence values. In Figure 3c and 3d, the red and black dashed lines outline a region dense with ridges, indicating a concentration of noisy data points in this area, which further highlights the complexity of the CIFAR-10 compared to MNIST. The trends of predicted ridges across different confidence intervals further validate the proposed predicted ridge confidence as shown in Figure 4. As the confidence intervals increase, the predicted ridge accuracy $A_{\mathrm{ridge}} = \frac{|\mathcal{B}_{\mathrm{pred}} \cap \mathcal{B}_{\mathrm{gt}}|}{|\mathcal{B}_{\mathrm{pred}}|}$ consistently improves over each round iterations. Moreover, as the model's performance improves, the number of high-confidence predicted ridges also increases.

## 4 VISUALIZATION OF SELECTION STRATEGIES IN ACTIVE LEARNING

To evaluate and compare different query strategies, we conducted a series of visualization experiments and analyses based on our proposed confidence decision boundary.

To facilitate better observation and comparison of strategies throughout the AL process, we designed three types of visualizations for the following experiments: **Confidence Decision Boundary**, **Cumulative Sampling over 5 Rounds**, and **Error Detection**, which are corresponding to the first,

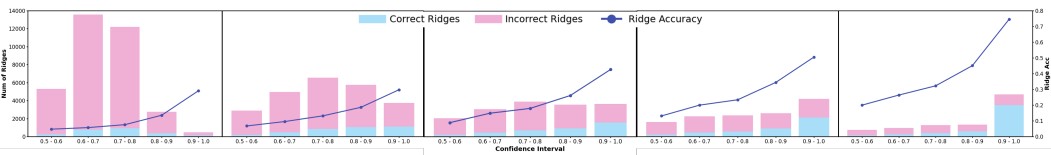

Figure 4: The visualization illustrates the trends in the total number of predicted ridges, the number of correct and incorrect ridges, and the predicted ridge accuracy for each confidence interval (CI) as model performance incrementally improves over round iterations from left to right.

second, and third rows of Figure 5, respectively. Each visualization serves a specific purpose: the Confidence Decision Boundary reflects the model's understanding of the entire dataset at a given stage, Cumulative Sampling highlights the model's selections under different strategies across multiple rounds, and Error Detection examines how these selections address the model's misconceptions. Furthermore, the maximum number of training samples accounted for only 1.22% of the total pool, and for CIFAR-10, this proportion was 1.225%. Consequently, we approximate the entire pool dataset as an extended test set to generate these three types of visualizations, allowing us to better illustrate the involved model's decision boundary, the characteristics of queried data, and the impact of newly sampled points on model performance.

The Confidence Decision Boundary, shown in the first row of Figure 5, is generated based on the model's predictions. For each ridge, we plot those where the representative points on either side have different predicted outcomes. The ridges are colored according to the predicted ridge confidence, with higher confidence indicating a greater likelihood that the points in the Voronoi cells on either side of the ridge belong to different classes. The Cumulative Sampling over 5 Rounds, shown in the second row of the Figure 5, illustrates the sampling process over multiple rounds. Each Voronoi cell is colored according to the true label of the representative data, and the sampled points for each round are marked with distinct colors. Additionally, each sampling point is annotated with its true label. The Errors Detection visualization, shown in the last row of the figure, highlights the model's error patterns after each training round. Red pentagrams indicate newly sampled data points that were added to the training set. The background reflects error regions based on whether the representative points were correctly predicted after training. Blue regions represent areas where the model made errors in the previous round that remain unresolved in the current round. Green regions indicate areas where the model corrected errors from the previous round. Purple regions highlight new errors made by the model in the current round, while blank regions represent areas where the model has already made correct predictions. The black dashed lines represent the ground truth boundary, providing a reference to observe how the clustering of true data and error regions evolves as the model's performance improves.

## 4.1 VISUALIZATION OF ENTROPY-BASED METHODS

Entropy is a key metric for measuring unpredictability in predicted class probabilities, widely used to quantify uncertainty in classification tasks. The three uncertainty methods discussed below all derive their original uncertainty information from entropy.

Entropy sampling directly estimates the uncertainty of samples based on a single set of model parameters, and its formula can be defined as:

$$H(x) = -\sum_{i=1}^{C} p(y_i|x) \log p(y_i|x) \tag{4}$$

where $p(y_i|x)$ is the predicted probability of class $y_i$, and $C$ is the number of classes.

MC Dropout effectively broadens the focus of traditional uncertainty methods, which primarily concentrate on predictive uncertainty. By performing multiple stochastic forward passes, the predictive entropy is calculated over averaged predictions:

$$H_{\mathrm{MC}}(x) = -\sum_{i=1}^{C} \left( \frac{1}{T} \sum_{t=1}^{T} p_t(y_i|x) \right) \log \left( \frac{1}{T} \sum_{t=1}^{T} p_t(y_i|x) \right) \tag{5}$$

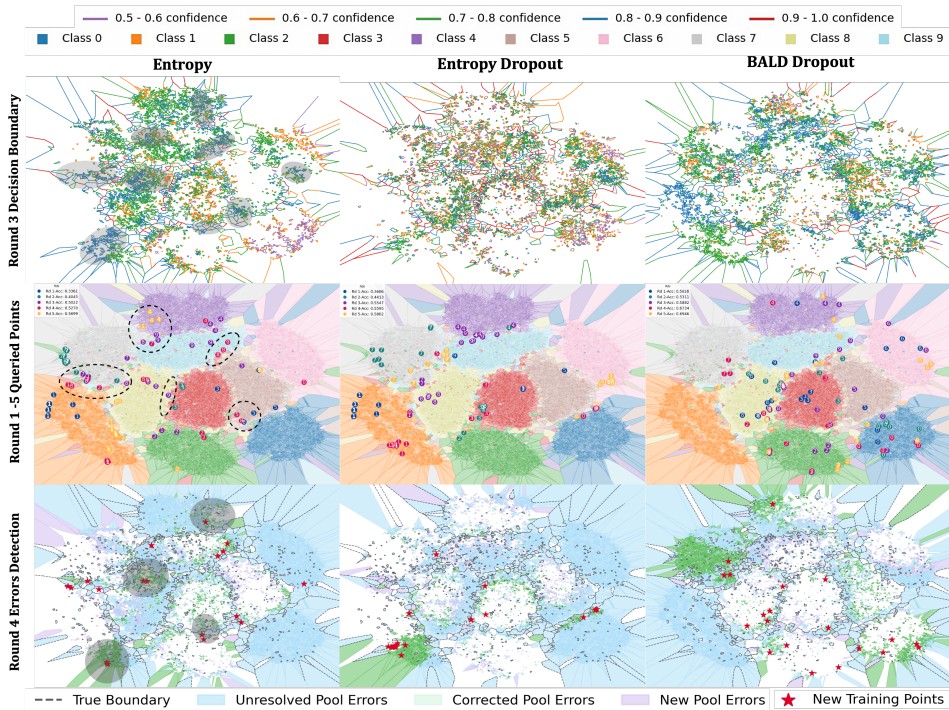

Figure 5: The first rows are the decision boundaries of models after three rounds of training, employing three different entropy-based methods. On the second row, we depict the distribution of sampled data from rounds 1 to 5 for each method with the class of each instance noted. The third row illustrates the impact of newly learned samples from the fourth round on the model.

where $T$ is the number of forward passes, and $p_t(y_i|x)$ is the predicted probability during the $t$-th pass.

The mutual information measured by BALD Dropout can be further decomposed as the difference between the predictive entropy and the expected conditional entropy:

$$I(x) = H_{\text{MC}}(x) - \frac{1}{T} \sum_{t=1}^{T} H_t(x) \tag{6}$$

This method selects points with maximum information gain about the model parameters from observing the label $y$.

Theoretically, these three methods exhibit a progressively layered structure, and our experiments reflect this. In Figure 5, illustrating the decision boundary from the third round of the model using the Entropy sampling strategy, we identified nine ambiguous regions based on the locations of the samples selected for the fourth round. A common feature of these regions is the presence of numerous high-confidence predicted ridges (in the current figure, the blue ridges are considered high-confidence predicted ridges) at the intersection of multiple classes (e.g., the regions between classes 1, 7, and 8). This visualization supports the concept by Settles (2009), where uncertainty methods select points near the decision boundary, with entropy sampling targeting points closer to high-confidence regions. A similar pattern was observed with entropy dropout. However, unlike entropy and entropy dropout, BALD dropout does not focus sampling as heavily in high-confidence predicted ridge regions but instead distributes the sampling more broadly across the regions.

In the visualization of cumulative sampling over the first five rounds in the second row of Figure 5, we observed that entropy sampling frequently engages in what we term "high-risk boundary-crossing" sampling. This behavior is characterized by selecting a small number of outlier data points located near the boundary of a minority class, while bypassing the boundary of a more populous class. The regions circled in black dashed lines indicate the areas where entropy sampling selected

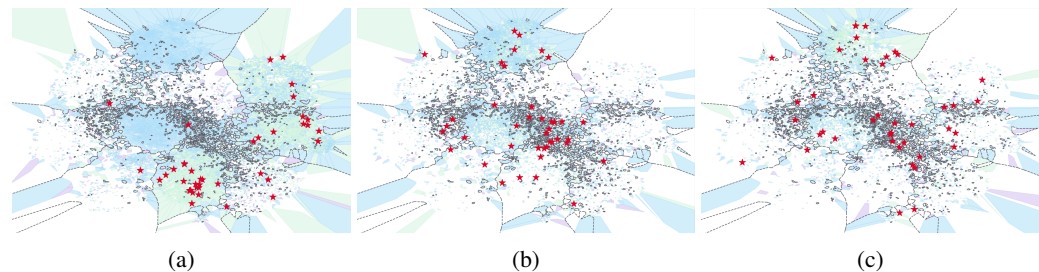

(a)           (b)           (c)

Figure 6: (a), (b), and (c) show the error detection visualizations for the model using Entropy Dropout sampling on CIFAR-10 at different rounds. (a) represents the 3rd round, (b) the 6th round, and (c) the 7th round.

such outlier points in the first five rounds. Notably, combining entropy with MC Dropout or Bayesian Convolutional Neural Networks can partially mitigate the "boundary-crossing" behavior.

However, by comparing the third-round sampling points in the cumulative sampling over 5 rounds and errors detection of Figure 5, we can observe that in the errors detection of entropy, the gray-shaded regions show how newly sampled points (red pentagrams) help correct large surrounding error areas (green). These points mainly come from the previous round's entropy sampling and are not located at multi-class intersections. A similar pattern occurs with entropy dropout, with some overlap in sampled points. In contrast, BALD dropout's broader sampling improves early learning efficiency compared to entropy and entropy dropout.

By leveraging Entropy Dropout, which results in concentrated sampling points and accounts for model parameter uncertainty, we observe two main trends. First, in Figure 6a, the red pentagrams (newly sampled points) are located in densely clustered regions of three classes. The surrounding green regions indicate corrected errors, showing that when the model lacks knowledge about a class, concentrated sampling reduces uncertainty caused by insufficient training samples. Second, once most of the uncertainty due to insufficient data is resolved, the model attempts to address uncertainty in regions with noisy data. As shown in Figures 6b and 6c, which depict two consecutive errors detection rounds, the model continuously samples points near the noisy areas. However, the corrected green regions remain limited. This suggests that the uncertainty in these areas arises from the noisy data itself, which contains mixed features from multiple classes. As a result, learning from a few noisy points does not necessarily lead to correcting errors across the entire noisy region.

## 4.2 VISUALIZATION OF LEAST CONFIDENCE AND MARGIN

Least Confidence and Margin sampling, though early uncertainty-based methods, are now rarely used as baselines for new approaches. However, they performed well in our experiment. To explore the reason, we compared them with Entropy and Random sampling. Since Least Confidence and Margin are similar in approach and results, the visualizations focus on Margin sampling, with Least Confidence results provided in the Appendix (see Figure 10).

From the ground truth boundary of CIFAR-10, we observe a central band-like region containing a dense concentration of true ridges, which is highlighted in Figure 3c. We identified the shape of this region based on the clustering of true ridges on the Figure 3c and marked it in Figure 7. Based on the banded area in Figure 7, we compared the uncertainty trends of models using Entropy, Margin, and Random strategies at two accuracy levels. Since the Margin-based model showed rapid performance improvement in the early stages, reaching an accuracy of approximately 0.78, which represents a lower accuracy but with a clearer decision boundary, we set the low accuracy threshold around 0.78 for comparison. In contrast, the Random sampling model achieved the highest accuracy of around 0.89 in this set of experiments, so the high accuracy threshold is set around 0.89. Accordingly, we selected the remaining models for comparison based on these performance benchmarks.

In Figure 7a, when the model before training performance is around 0.78, most of the samples selected through Entropy sampling are concentrated within the banded region outlined by the green lines. The remaining samples are distributed in areas where high-confidence predicted ridges cluster in the confidence decision boundary of model before training, as well as in the error regions (blue and

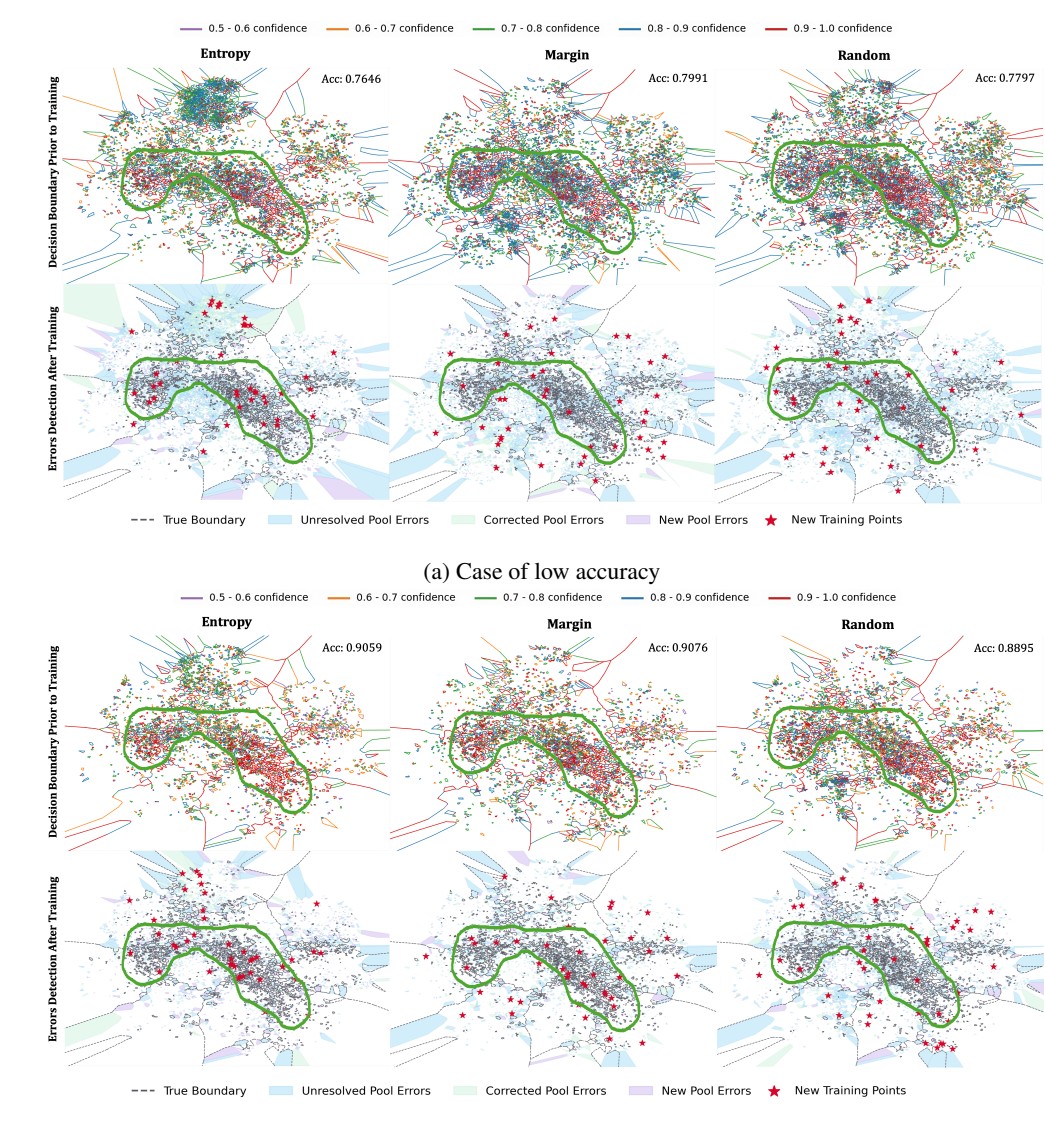

(a) Case of low accuracy

(b) Case of high accuracy

Figure 7: The first row in (a) and (b) depicts the decision boundary of the queried model across different confidence intervals on CIFAR-10, where regions with a higher density of high-confidence predicted ridges correspond to areas the model finds less familiar. The second row shows the impact on the model after incorporating the newly queried data into training, with the blue and green areas representing regions where the model made errors prior to training.

green) from the errors detection of model after training. In contrast, the sampling points for Margin are more dispersed, with fewer points located in noisy regions, and most of the points concentrated in the clusters of various classes. Margin at this stage tends to select high-uncertainty samples that are easier to resolve.

In Figure 7b, when the performance of model before training is at a relatively good level around 0.9, the confidence decision boundary obtained by the model is significantly clearer than in Figure 7a. The ridges outlining class boundaries at the periphery have been reduced to a few high-confidence predicted ridges, with more high-confidence predicted ridges now concentrated in the banded region. At this point, Entropy sampling still focuses on regions similar to those highlighted by green line in Figure 7a. However, the Margin sampling shows a significant shift compared to the low-accuracy case, with more samples now appearing in the noisy region, shifting from primarily sampling areas

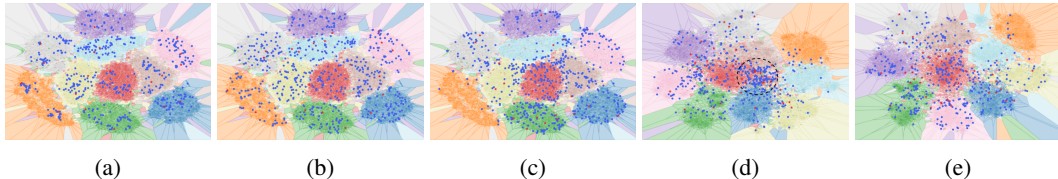

(a)  (b)  (c)  (d)  (e)

Figure 8: (a) KMeans on MNIST; (b) Random on MNIST; (c) KCenterGreedy on MNIST; (d) KCenterGreedy on CIFAR-10; (e) KCenterGreedy on CIFAR-10 drawn by the feature extracted from AL model

of uncertainty due to insufficient data to targeting regions where noisy points are concentrated. The proportion of these two types of uncertainty changes as the model's performance improves.

### 4.3 VISUALIZATION OF DIVERSITY-BASED METHOD

Diversity methods focus on reducing redundancy by selecting data points that cover a wide range of features or data distributions Brinker (2003), ensuring comprehensive input space coverage and improving model generalization. However, once diversity-based methods reach a certain performance threshold, further improvement becomes increasingly difficult without incorporating uncertainty estimation methods, as shown in Figure 7). In later stages, as the model gains a deeper understanding of the fundamental characteristics of each class, the remaining errors are primarily concentrated near the ground truth boundary (black dashed line) or in clusters of noisy points (band-link region outlined by the green lines). Since these regions occupy a small portion of the overall space, diversity-based methods have a low probability of successfully targeting these areas. Furthermore, different diversity methods distribute sampled points differently. As shown by the cumulative sampling points in Figure 8, k-means tends to cluster points in the center of each class, whereas random sampling evenly covers the plane. This broader coverage gives random sampling a higher chance of selecting points in these small error-making regions, thereby contributing to its consistently stable performance.

However, we observed that KCenterGreedy suffers from significant class imbalance during sampling. Figure 8e visualizes the dimensionality-reduced feature space extracted by the current model. Compared to the densely sampled central region in Figure 8d, the peripheral regions are more sparsely sampled, with some classes showing large areas of empty space. The more widespread and dispersed cumulative sampling points based on the model's own feature extraction suggest that KCenterGreedy is influenced by the model's biased understanding of the data distribution. Additionally, the high dimensionality of the feature space extracted by the model makes it difficult to accurately assess distance differences. This visualization revealing the sampling imbalance caused by KCenterGreedy's sensitivity to model bias further supports Yehuda et al. (2022), which found that KCenterGreedy performs poorly in multi-class tasks when the sampling budget is constrained.

## 5 CONCLUSION

In this work, we introduced a novel confidence decision boundary visualization method for AL, utilizing Voronoi tessellation to provide a more granular and informative representation of decision boundaries evaluated by a ridge confidence metric. This approach enables a deeper understanding of various AL query strategies by highlighting nuanced differences in how models perform sampling, handle uncertainty and respond according to sampled data. Our visualizations revealed important insights into the behavior of the strategies selected for this experiment, but their applicability extends beyond these specific methods. Notably, we observed two key trends in uncertainty: concentrated sampling effectively reduces uncertainty caused by insufficient training samples, while uncertainty in noisy regions is harder to resolve due to mixed class features. These findings emphasize the importance of selecting appropriate query strategies to handle different types of uncertainty and improve model performance. Overall, our approach provides a valuable tool for understanding and analyzing AL strategies, addressing the limitations of traditional visualizations.

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

# A    APPENDIX

The proposed algorithm leverages Voronoi tessellation and Delaunay triangulations to construct a confidence decision boundary. As shown in Algorithm 1, we first compute the Delaunay triangulation $\mathcal{D}(\mathcal{P})$ (Line 2) and determine the circumcenters of each triangle (Line 5), which form the vertices of Voronoi cells.

Each edge $e = (\boldsymbol{p}_i, \boldsymbol{p}_j)$ is examined to identify Voronoi ridges, and if $\hat{y}_i \neq \hat{y}_j$, the predicted ridge confidence $C_{\text{pred}}(e)$ is calculated using class probabilities (Line 14). The Voronoi ridge and its confidence are added to $\mathcal{B}_{\text{pred}}$, forming the overall confidence decision boundary $\mathcal{B}_{\text{pred}}$.

---

**Algorithm 1** Algorithm to Generate Confidence Decision Boundary

---

1: **Input**: Point set $\mathcal{P} = \{\boldsymbol{p}_1, \boldsymbol{p}_2, \ldots, \boldsymbol{p}_n\}$, predicted probabilities $\{P(\hat{y}_i = k) \mid k = 1, 2, \ldots, K, \ \forall \boldsymbol{p}_i \in \mathcal{P}\}$
2: Compute the Delaunay triangulation $\mathcal{D}(\mathcal{P})$ of the point set $\mathcal{P}$.
3: Initialize an empty dictionary circumcenters $\mathcal{C}$ to store the circumcenters of triangles.
4: **for** each triangle $t = \triangle(\boldsymbol{p}_i, \boldsymbol{p}_j, \boldsymbol{p}_k) \in \mathcal{D}(\mathcal{P})$ **do**
5:     Compute the circumcenter $\boldsymbol{c}_t = (U_x, U_y)$ of triangle $t$ using:

$$D = 2 \begin{vmatrix} x_i & y_i & 1 \\ x_j & y_j & 1 \\ x_k & y_k & 1 \end{vmatrix},$$

$$U_x = \frac{1}{D} \begin{vmatrix} x_i^2 + y_i^2 & y_i & 1 \\ x_j^2 + y_j^2 & y_j & 1 \\ x_k^2 + y_k^2 & y_k & 1 \end{vmatrix},$$

$$U_y = \frac{1}{D} \begin{vmatrix} x_i^2 + y_i^2 & x_i & 1 \\ x_j^2 + y_j^2 & x_j & 1 \\ x_k^2 + y_k^2 & x_k & 1 \end{vmatrix}.$$

6:     Store $\boldsymbol{c}_t$ in the dictionary: circumcenters $\mathcal{C}[t] = \boldsymbol{c}_t$.
7: **end for**
8: Initialize an empty set $\mathcal{B}_{\text{pred}}$ to store the confidence decision boundary.
9: **for** each edge $e = (\boldsymbol{p}_i, \boldsymbol{p}_j)$ in the Delaunay triangulation $\mathcal{D}(\mathcal{P})$ **do**
10:     Find the two triangles $t_1$ and $t_2$ that share edge $e$.
11:     Retrieve the corresponding circumcenters $\boldsymbol{c}_{t_1}$ and $\boldsymbol{c}_{t_2}$.
12:     Construct the corresponding Voronoi ridge $e_V = (\boldsymbol{c}_{t_1}, \boldsymbol{c}_{t_2})$.
13:     **if** $\hat{y}_i \neq \hat{y}_j$ **then**
14:         Compute the predicted ridge confidence $C_{\text{pred}}(e)$:

$$C_{\text{pred}}(e) = 1 - \sum_{k=1}^{K} P(\hat{y}_i = k) \times P(\hat{y}_j = k)$$

15:         Add $(e_V, C_{\text{pred}}(e))$ to the set $\mathcal{B}_{\text{pred}}$.
16:     **end if**
17: **end for**
18: **Return** the confidence decision boundary $\mathcal{B}_{\text{pred}}$.

---

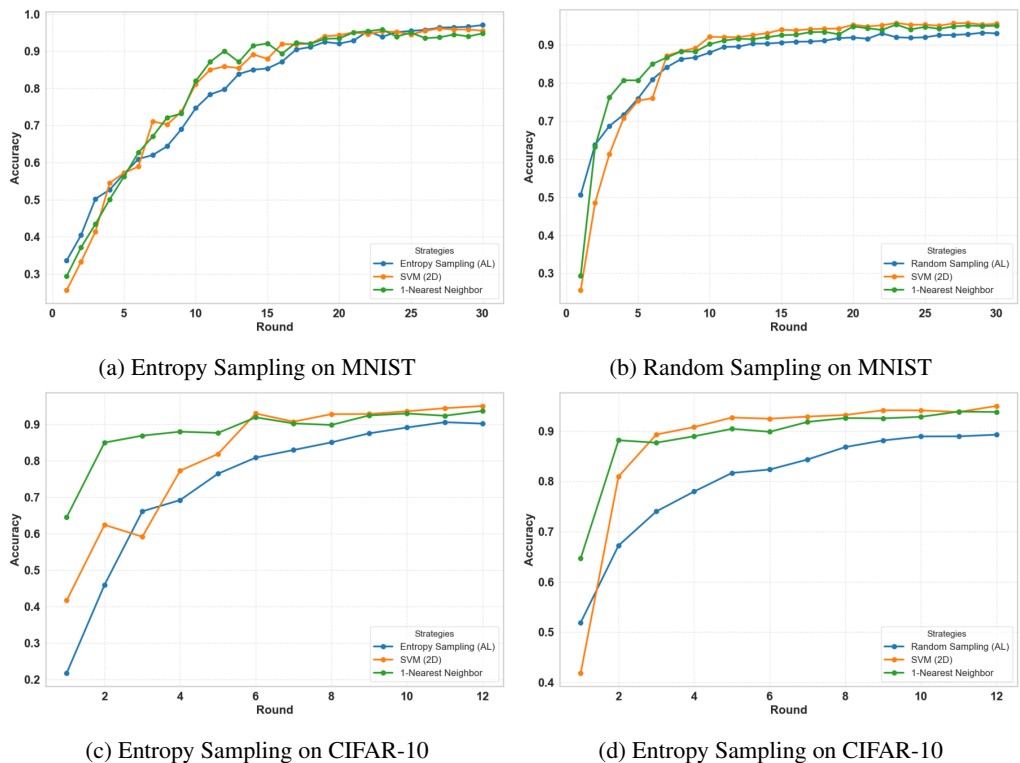

Figure 9: Results of Local Structure Preservation: We trained and tested an SVM on the same training data as in each round of the AL process, but used the corresponding dimensionality-reduced 2D features for visualization instead of the original data, and compared its accuracy with that of the model in the AL process trained on the original data. Additionally, we evaluated the results of a 1-NN classifier on the 2D test set.

| Pearson Correlation↑ | EntropySampling (Avg) | RandomSampling (Avg) | Visualization Model |
|:---:|:---:|:---:|:---:|
| **MNIST** | 0.6307 | 0.6630 | 0.5951 |
| **CIFAR-10** | 0.5049 | 0.4615 | 0.5166 |

Table 1: Comparison of Pearson Correlation across Sampling Strategies: This method evaluates the Pearson correlation between the pairwise similarity of the features extracted by the model and the pairwise distance matrix calculated from the dimensionality-reduced data.

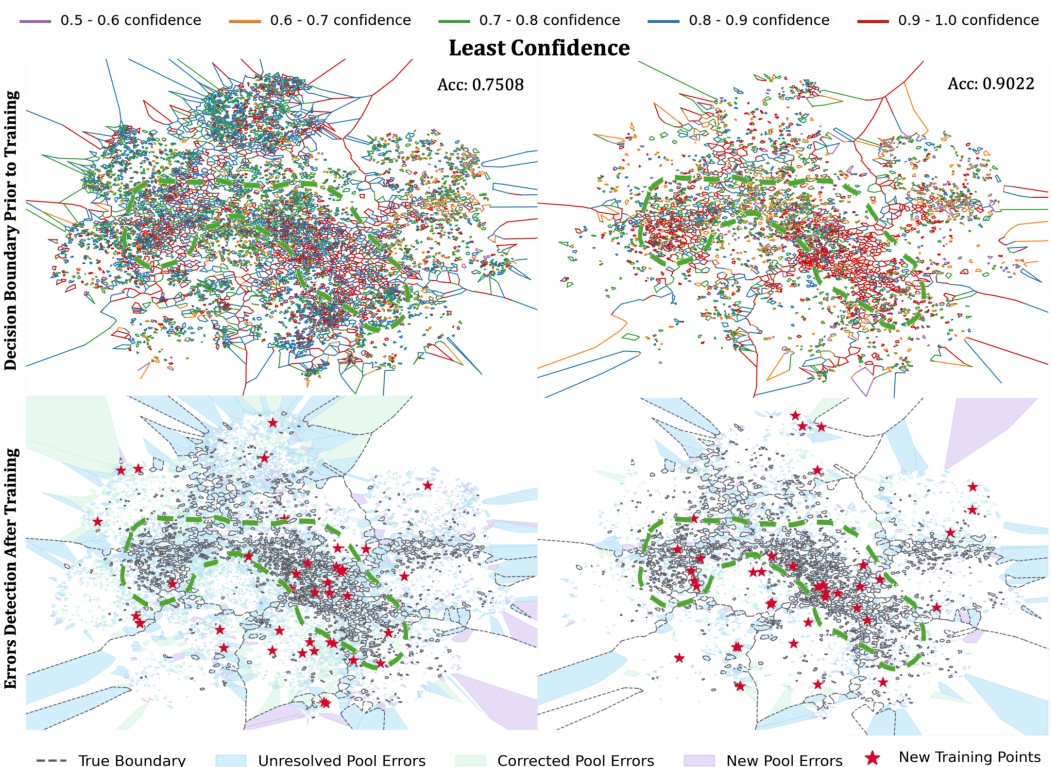

Figure 10: The first row in depicts the decision boundary of the Least Confidence based-model of different accuracy across different confidence intervals on CIFAR-10, where regions with a higher density of high-confidence predicted ridges correspond to areas the model finds less familiar. The second row shows the impact on the model after incorporating the newly queried data into training, with the blue and green areas representing regions where the model made errors prior to training.

Figures 11-15 illustrate the visualization application of the **Entropy Sampling Dropout** strategy on the CIFAR-10 dataset. Figure 11 and Figure 12 compare the evolution of confidence decision boundaries on dynamically updated features obtained from the AL model in each round and fixed features generated from a visualization model, respectively. These decision boundaries are depicted across initial round 0 and 12 rounds of the AL process, showcasing how the feature space adapts iteratively as new samples are incorporated into the model. Similarly, Figure 13 and Figure 14 focus on error detection over the same rounds, contrasting dynamically updated features during rounds with fixed visualization features to emphasize differences in error propagation and resolution during the AL process. Lastly, Figure 15 highlights iterative sampling trends over the rounds, summarizing how selected samples compound to shape the training dataset. Together, these figures offer a comprehensive view of how the AL framework evolves through iterative sampling and model refinement.

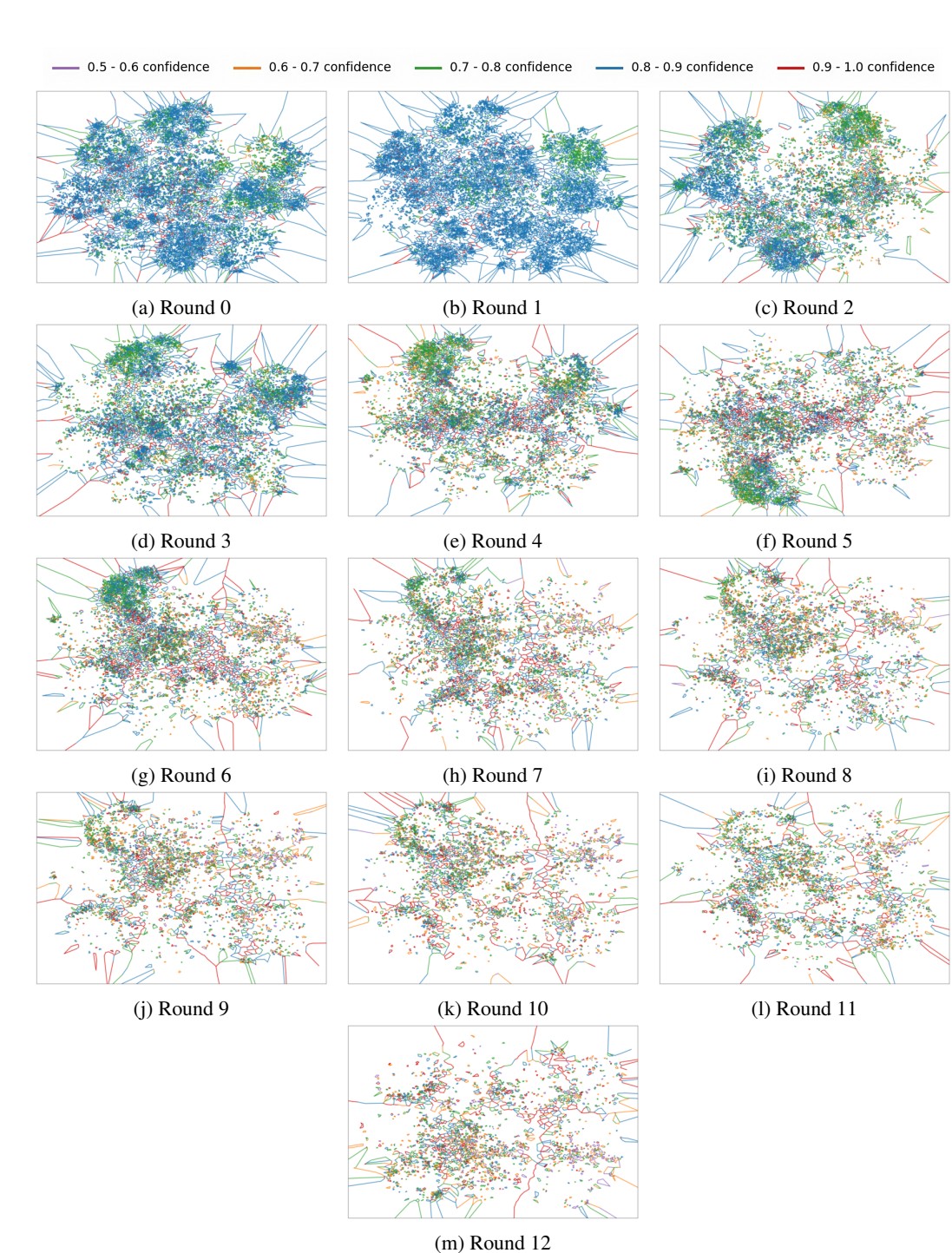

Figure 11: Confidence Decision Boundary on dynamically updated features obtained from AL model in each round

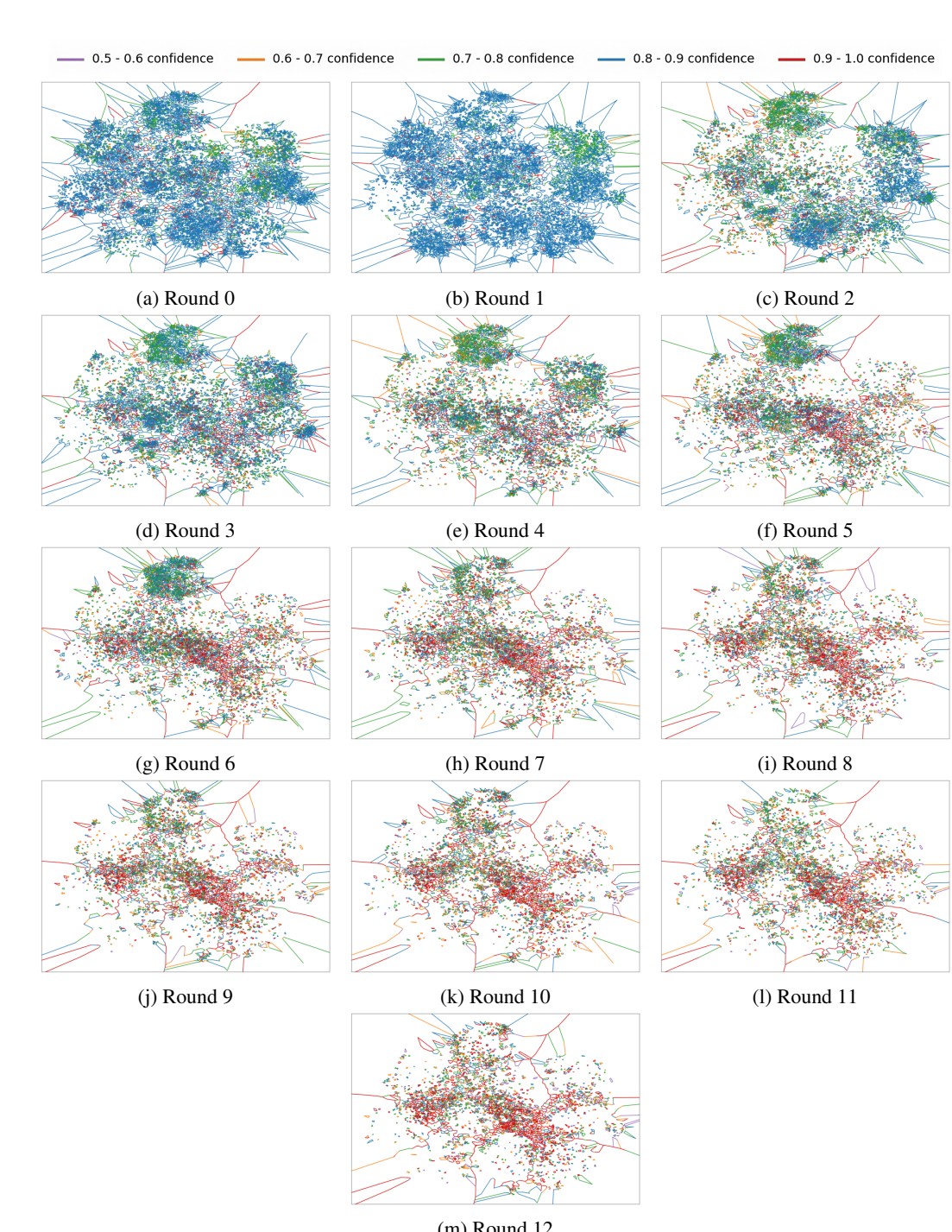

Figure 12: Confidence Decision Boundary on fixed features obtained from visualization model for posterior analysis

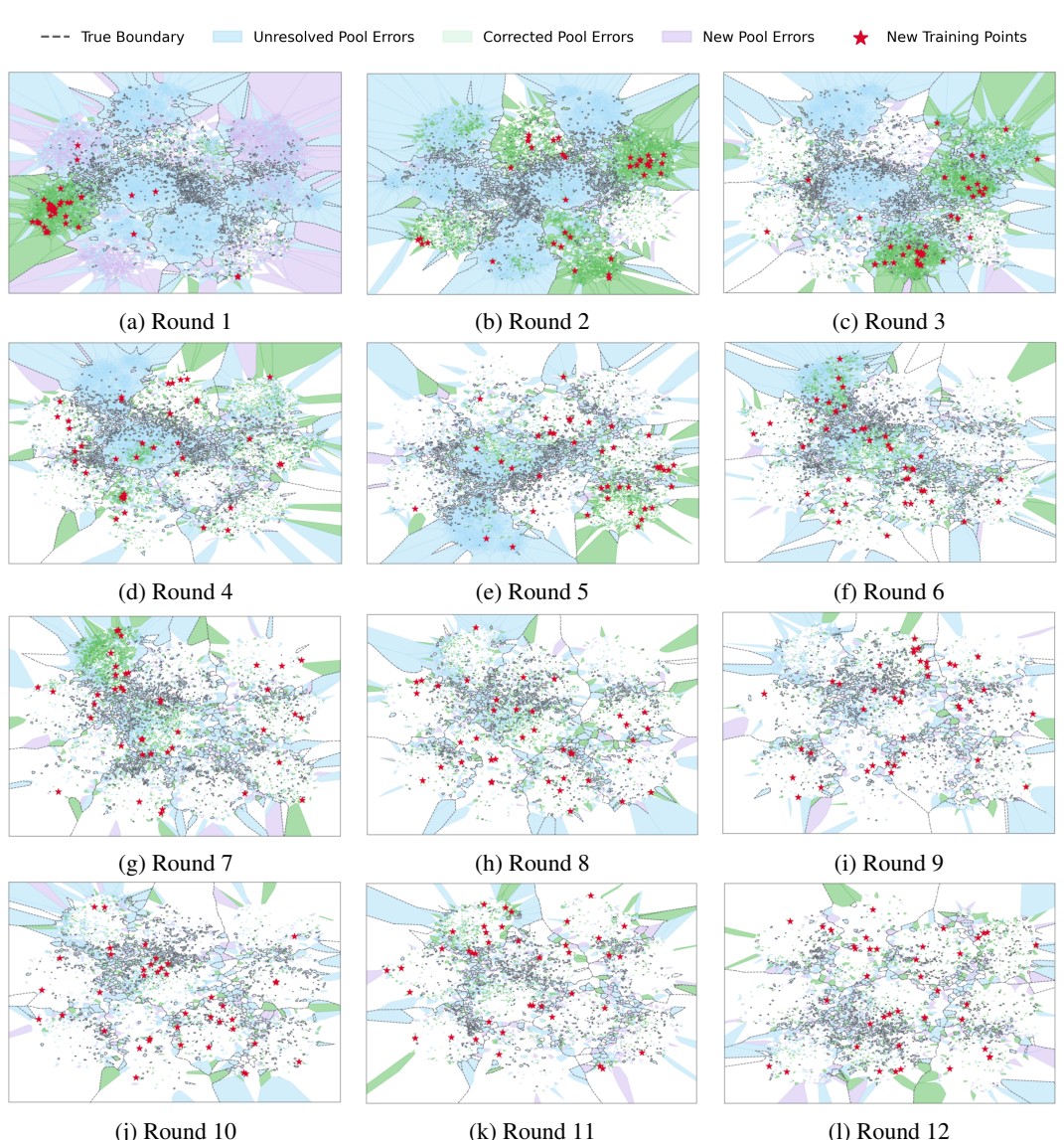

Figure 13: Error Detection on dynamically updated features obtained from AL model in each round

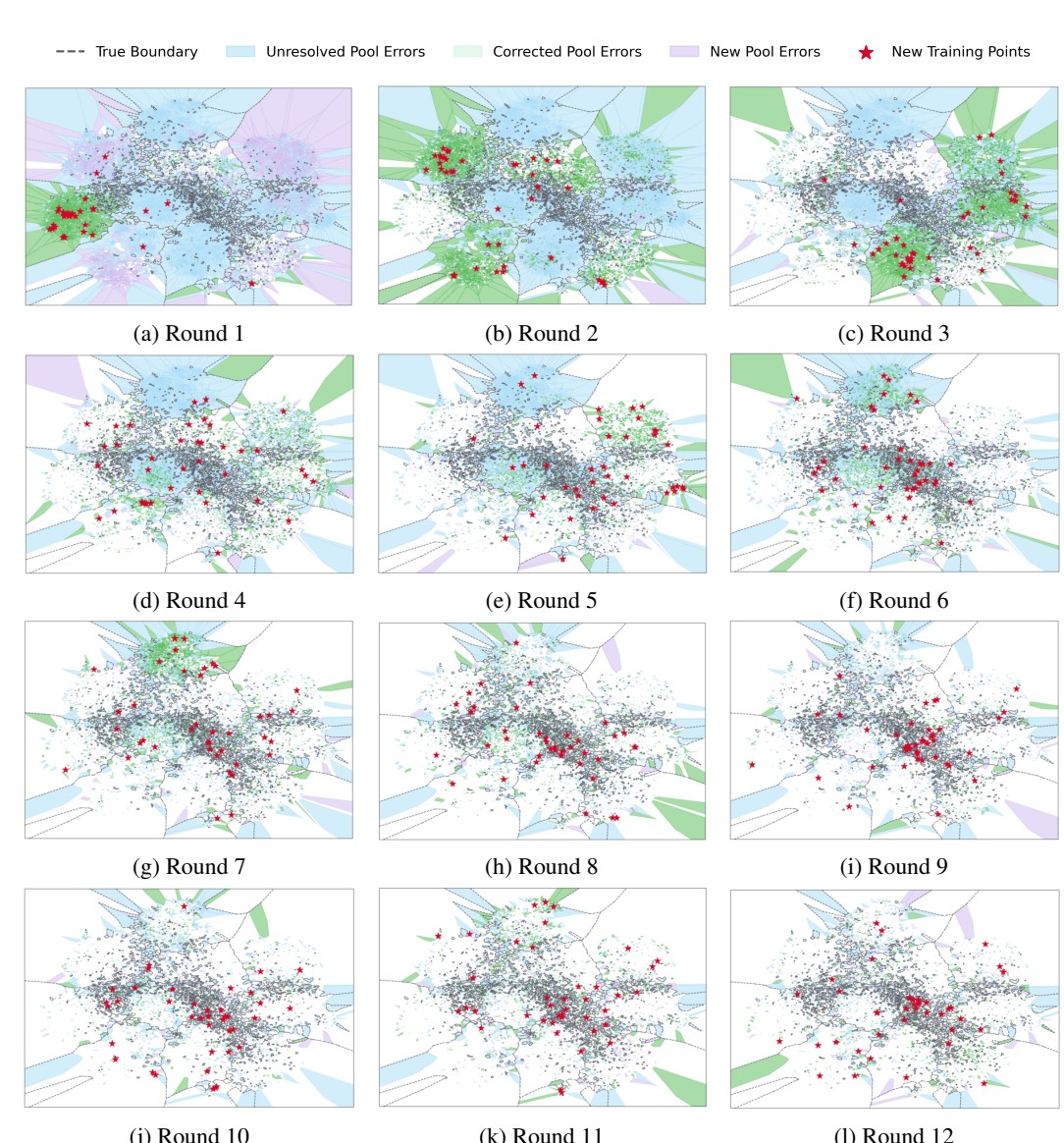

Figure 14: Error Detection on fixed features obtained from visualization model for posterior analysis

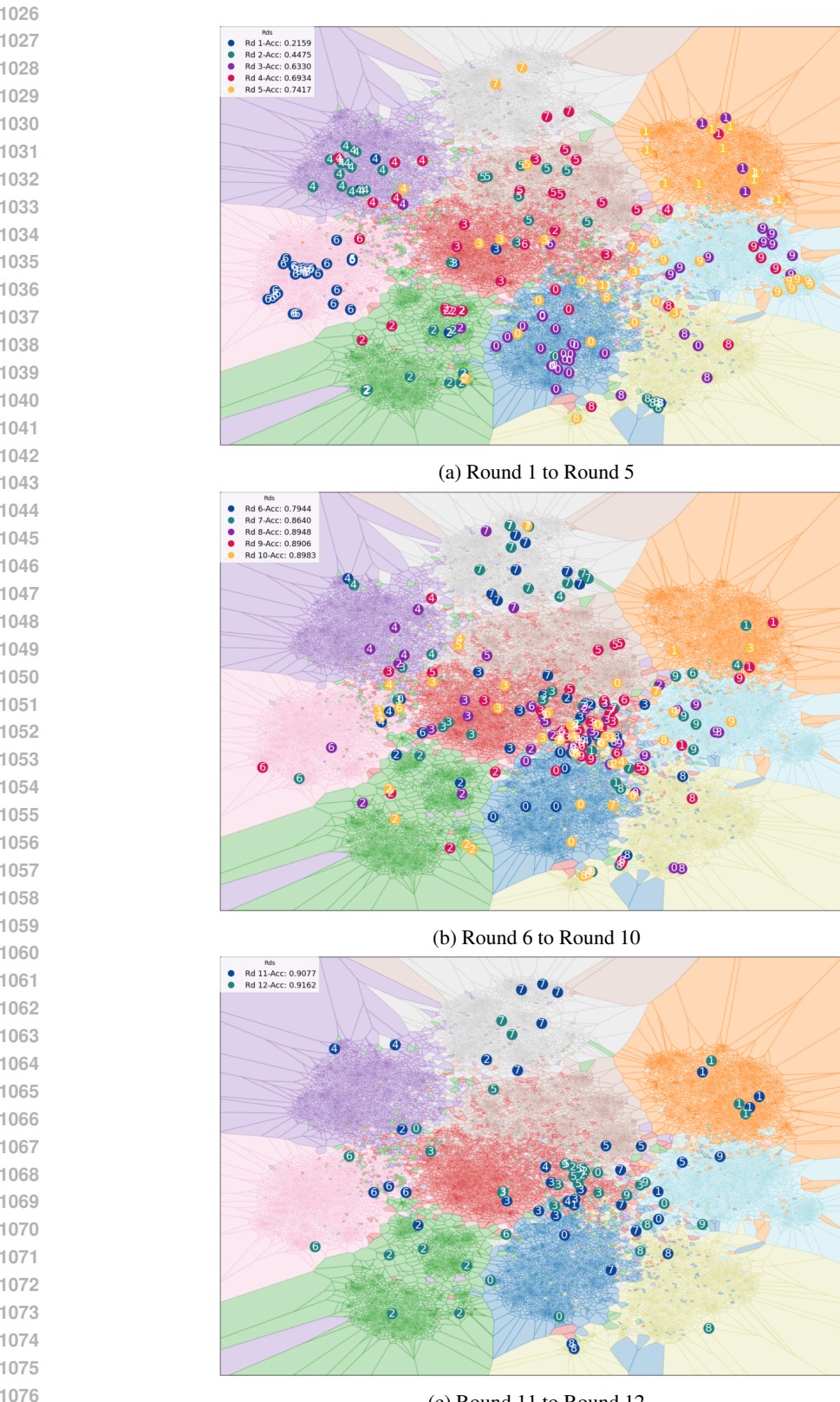

(a) Round 1 to Round 5

(b) Round 6 to Round 10

(c) Round 11 to Round 12

Figure 15: Cumulative Sampling on fixed features obtained from visualization model for posterior analysis

