# OpenReview forum: "Voronoi Tessellation-based Confidence Decision Boundary Visualization to Enhance Understanding of Active Learning"
_ICLR.cc/2025/Conference — Submitted to ICLR 2025_

### Official Review · Reviewer_rwsQ · 2024-11-02

**Soundness:** 2
**Presentation:** 3
**Contribution:** 2
**Rating:** 5
**Confidence:** 4

**Summary:**

This paper presents a Voronoi tessellation-based method to visualize decision boundaries in active learning. By highlighting ridges between cells with differing predictions, this method provides a clearer view of the decision boundaries. In addition, it introduces a ridge confidence metric to quantify prediction uncertainty. Experiments on MNIST and CIFAR-10 illustrate the effectiveness of this approach for analyzing various active learning strategies, providing insights into the behavior and effectiveness of different sampling techniques.

**Strengths:**

1. Comprehensive Comparison of Active Learning Strategies: The evaluation includes a systematic comparison of various active learning strategies. These potentially contribute to the broader discourse on active learning methods and offer practical guidance for future research in this area.
2. Effective Use of Visualization Techniques: The paper incorporates visualization methods to facilitate the analysis of data and models. This makes the analysis and findings more accessible and intuitive.

**Weaknesses:**

1. **Concerns about Decision Boundary Generation**. From the figures, the projection results and the Voronoi tesselation results appear to be fixed across multiple rounds. However, as feature vectors update during training, the projection results and the decision boundaries should also be updated. It is essential to clarify how well these fixed tessellations with the updated predictions capture the model's behavior. In addition, the use of 2D projections to represent high-dimensional decision boundaries raises concerns, as results can vary significantly based on the selected projection method and parameters.
2. **Insufficient evaluation**. While this paper compares different active learning strategies and summarizes some insights, this evaluation is not sufficient and rigorous. On the one hand, the evaluation is only conducted on MNIST and CIFAR-10, which only contain a few classes with substantial differences between them. It remains uncertain how the proposed method performs with more classes or finer distinctions. On the other hand, the evaluation of boundary effectiveness is inadequate. Many insights, such as oversampling on the boundary region in the uncertainty-based methods, can be identified from scatterplots alone, making the Voronoi approach seem unnecessary.
3. **Omission of Relevant Literature**. The paper overlooks significant works that utilize Voronoi tessellation for visualizing sample boundaries. For instance, Chen et al. [*] use it to visualize the samples and boundaries in the analysis of label propagation. Including such references would enhance the contextual foundation of the research.
[*] Interactive Graph Construction for Graph-Based Semi-Supervised Learning. TVCG 2021.

**Questions:**

1. Decision Boundary Representation: How do the fixed Voronoi tessellation results relate to the updated feature vectors during training? Can you explain how this approach ensures an accurate representation of the model's decision boundaries throughout the training process?
2. Generalizability: How does the method work on more complicated datasets, such as ImageNet (with more classes) and CUB-200-2011 (fine-grained classification tasks)
2. Boundary Effectiveness: Can you elaborate on how the Voronoi tessellation approach adds value beyond insights obtainable from scatterplots? What specific contributions does it provide in terms of understanding boundary effectiveness that traditional methods do not?

---

> ### Author Response · Authors · 2024-11-26
>
> We sincerely appreciate the reviewer **rwsQ** for the comment.
>
> ### **Q1: How do the fixed Voronoi tessellation results relate to the updated feature vectors during training?**
>
> There is a misunderstanding regarding the features we used to generate the figures. In lines 208-232 of the paper, we describe the features used to generate the Voronoi diagram. In fact, we have two ways of constructing a 2D feature map:
>
> 1. **Dynamically Updated Features**: These features are dynamically updated during active learning (e.g., Figure 2). They are extracted by the trained model at each round on the pool data and test data, with the dimensionality-reduced results serving as Voronoi points for constructing our visualization.
>
> 2. **Fixed Features for Posterior Analysis**: These features are used for posterior analysis (e.g., Figures 5, 6, and 7). Because we need to observe and compare the sampling behaviors and results of different query strategies, we focus on analyzing the confidence decision boundary, sampling points (newly queried points), and the impact of new samples on model training (error detection) using a fixed feature map. This approach is necessary due to differences introduced by the features extracted by models from different rounds and further variations caused by different dimensionality reduction methods.
>
> Thus, the second method involves training a **visualization model** (described in Section 3.1.3) with all the pool data to help generate fixed features, which are then used for the feature map in every round. The reason for using the visualization model is explained in lines 211-215 of the paper.
>
>
> ### **Q2: How does the method work on more complicated datasets?**
>
> We believe our method can works well on large-scale datasets. If the dataset is exceedingly large, a subset of it can be used for visualization. On a smaller dataset that is independently and identically distributed (i.i.d.) with the larger dataset, the decision boundary of the trained model shows relatively small differences compared to the one trained on the larger dataset. We computed the prediction differences on the test set and the overlap of decision boundaries (formed by predicted ridges) on the test set between models trained on smaller datasets of varying sizes and the **Visualization Model (VM)** trained on the whole pool dataset. The experimental results are as follows:
>
> #### - Prediction Differences:
>
> | Model trained on                  | KL Divergence↓ | JS Divergence↓ | Wasserstein Distance↓ |
> |--------------------------|----------------|----------------|------------------------|
> | **MNIST (1000 samples)** | 0.1890         | 0.0324         | 0.0067                |
> | **MNIST (5000 samples)** | 0.0515         | 0.0101         | 0.0028                |
> | **MNIST (10000 samples)**| 0.0362         | 0.0067         | 0.0019                |
> | **CIFAR-10 (1000 samples)** | 0.1255      | 0.0318         | 0.0121                |
> | **CIFAR-10 (5000 samples)** | 0.0371      | 0.0094         | 0.0045                |
> | **CIFAR-10 (10000 samples)**| 0.0177      | 0.0043         | 0.0025                |
>
> The smaller the values of KL Divergence, JS Divergence, and Wasserstein Distance, the more similar the two predicted probability distributions are.
>
> #### - Overlapping Predicted Ridges:
>
>
> | Model trained on                  | Total Predicted Ridges↓ | Overlapping Predicted Ridges (compared to VM)↑ |
> |--------------------------|----------------|--------------------------------------|
> | **CIFAR-10 (1000 samples)** | 3728         | 2587                                 |
> | **CIFAR-10 (5000 samples)** | 3479         | 3072                                 |
> | **CIFAR-10 (10000 samples)**| 3520         | 3233                                 |
> | **CIFAR-10 (40000 samples) VM** | 3426      | 3426                                 |
>
>
> | Model trained on                  | Total Predicted Ridges↓ | Overlapping Predicted Ridges (compared to VM)↑ |
> |--------------------------|----------------|--------------------------------------|
> | **MNIST (1000 samples)** | 2919          | 1224                                 |
> | **MNIST (5000 samples)** | 1936          | 1435                                 |
> | **MNIST (10000 samples)**| 1880          | 1518                                 |
> | **MNIST (50000 samples) VM** | 1799       | 1799                                 |
>
> Additionally, we will fill different cells with colors to make it easier to observe the changes in regions.
>
> The real issue is likely to arise when there are many classes. Anything beyond approximately 20 classes becomes very challenging to visualize. To address this, we recommend splitting the task into multiple sub-classification tasks and performing visualization on these smaller subsets.
>
> Our approach mainly focuses on exploring the sampling behaviors of classical query strategies and analyzing the impact of newly sampled points on the model’s training in the subsequent round.

---

> > ### Author Response · Authors · 2024-11-26
> >
> > ### **Q3: What specific contributions does it provide in terms of understanding boundary effectiveness that traditional methods do not?**
> >
> > There are very few visualization methods specifically designed for active learning (AL). Most existing approaches provide only basic visualizations accompanying the proposal of certain query strategies, such as visualizing the distribution of queried samples for comparison or explanation purposes [1, 2, 3]. Alternatively, some visualizations depict the impact of training samples on the decision boundary of simple models to demonstrate the sampling process and particular scenarios [4].
> >
> > Similarly, there are also not many decision boundary visualization methods with general applicability. For example, in active learning, existing methods often provide only the changes in the decision boundary over time but fail to jointly reflect the relationships between the training data and the decision boundary, as well as the connections between the decision boundary and the sampling data selected by the model. In contrast, our method can visualize these relationships, enabling a deeper investigation into active learning query strategies.
> >
> > Moreover, while dimensionality reduction for visualization inevitably introduces spatial distortion, our proposed confidence decision boundary captures the relationship between the predicted ridges (which collectively form the decision boundary) in the 2D visualization and their corresponding classified points in the high-dimensional space. Unlike traditional approaches that treat the decision boundary uniformly, our method incorporates ridge confidence to highlight the varying levels of complexity and uncertainty in different regions of the boundary.
> >
> > #### References
> >
> > [1] Agarwal, Sharat, et al. "Contextual diversity for active learning." *Computer Vision–ECCV 2020: 16th European Conference, Glasgow, UK, August 23–28, 2020, Proceedings, Part XVI 16.* Springer International Publishing, 2020.
> > [2] Pinsler, Robert, et al. "Bayesian batch active learning as sparse subset approximation." *Advances in neural information processing systems 32* (2019).
> > [3] Liu, Zhuoming, et al. "Influence selection for active learning." *Proceedings of the IEEE/CVF international conference on computer vision.* 2021.
> > [4] Tharwat, Alaa, and Wolfram Schenck. "A survey on active learning: State-of-the-art, practical challenges and research directions." *Mathematics* 11.4 (2023): 820.
> >
> >
> > ### **Q4: Omission of relevant literature**
> >
> > We appreciate you pointing out the omission of relevant literature. While the referenced paper does use Voronoi, it does not apply it to AL, which is why we initially did not consider it.
> >
> > In the revision, we will include this reference and discuss its connection to our approach.

---

> > > ### Comment · Reviewer_rwsQ · 2024-11-26
> > >
> > > I appreciate the authors' feedback and the work they've put into addressing my concerns. Although I’ve increased my grade from 3 to 5, my concerns about using 2D projections to represent high-dimensional decision boundaries still persist. I would be grateful if the authors could further discuss this aspect.

---

> ### Author Response · Authors · 2024-11-27
>
> We really appreciate reviewer **rwsQ's** response. We believe your concern is very insightful, especially in the context of low-dimensional visualizations.
>
> Different dimensionality reduction methods result in varying degrees of information loss, which can cause spatial distortion in our visualization results and, consequently, affect the decision boundary constructed in the reduced dimensions. However, this spatial distortion is an inherent and objective issue, and its impact is limited according to the experimental results. Our visualization method does not rely on any specific dimensionality reduction technique, and exploring the effect caused by different dimensionality reduction methods across various datasets is not the core focus of this work. However, in the revision, we will include an analysis of the effect caused by the dimensionality reduction technique currently employed (t-SNE) in our method.
>
> We used two different methods for evaluation:
>
> 1. **Correlation Test** [1, 2]:
>    This method evaluates the Pearson correlation between the pairwise similarity of the features extracted by the model and the pairwise distance matrix calculated from the dimensionality-reduced data. The results are shown in the table below:
>
>    | &#124;Pearson Correlation&#124;↑     | EntropySampling (Avg) | RandomSampling (Avg) | Visualization Model |
>    |--------------------------|-----------------------|-----------------------|---------------------|
>    | **MNIST**               | 0.6307               | 0.6630               | 0.5951             |
>    | **CIFAR-10**            | 0.5049               | 0.4615               | 0.5166             |
>
>    Since sampling occurs over multiple rounds, we calculated the average values. The absolute value of Pearson Correlation closer to 1 indicates a higher correlation, while values closer to 0 indicate no correlation. The results indicate that the impact of spatial distortion caused by dimensionality reduction is relatively small.
>
> 2. **Local Structure Preservation** [3]:
>    We trained and tested an SVM on the same training data as in each round of the AL process, but used the corresponding dimensionality-reduced 2D features for visualization instead of the original data, and compared its accuracy with that of the model in the AL process trained on the original data. Additionally, we evaluated the results of a 1-NN classifier on the 2D test set. The results are as follows:
> #### - Accuracy:
>    | Round | AL RandomSampling (CIFAR-10) | SVM     | 1-NN    |
>    |-------|-------------------------------|---------|---------|
>    | 0     | 0.3399                        | 0.4186  | 0.6467  |
>    | 1     | 0.5191                        | 0.8096  | 0.8817  |
>    | 2     | 0.6722                        | 0.8931  | 0.8768  |
>    | 3     | 0.7401                        | 0.9080  | 0.8897  |
>    | 4     | 0.7797                        | 0.9269  | 0.9044  |
>    | 5     | 0.8167                        | 0.9246  | 0.8985  |
>    | 6     | 0.8235                        | 0.9288  | 0.9182  |
>    | 7     | 0.8434                        | 0.9320  | 0.9259  |
>    | 8     | 0.8682                        | 0.9414  | 0.9253  |
>    | 9     | 0.8814                        | 0.9411  | 0.9283  |
>    | 10    | 0.8893                        | 0.9380  | 0.9388  |
>    | 11    | 0.8895                        | 0.9499  | 0.9379  |
>
>    It can be observed that both SVM and 1-NN achieve better performance compared to the AL process, indicating that the dimensionality reduction method effectively preserves the local structural characteristics of the original data. More experimental results will be included in the appendix.
>
> #### References
>
> [1] Smyth, Barry, Mark Mullins, and Elizabeth McKenna. "Picture Perfect: Visualisation Techniques for Case-based Reasoning." *ECAI*. 2000.
> [2] Namee, Brian Mac, and Sarah Jane Delany. "CBTV: Visualising case bases for similarity measure design and selection." *International conference on case-based reasoning*. Berlin, Heidelberg: Springer Berlin Heidelberg, 2010.
> [3] Huang, Haiyang, et al. "Towards a comprehensive evaluation of dimension reduction methods for transcriptomic data visualization." *Communications Biology* 5.1 (2022): 719.

---

### Official Review · Reviewer_8SHQ · 2024-11-02

**Soundness:** 3
**Presentation:** 3
**Contribution:** 3
**Rating:** 6
**Confidence:** 3

**Summary:**

This paper introduces the confidence decision boundary visualization method, which provides an interpretable comparison of various active learning strategies, yielding valuable insights. The experimental design and discussion are thorough.
I’m glad to see that some work at the ICLR conference focuses on data visualization. The insights provided by data visualization can better help users understand some of the underlying aspects behind the models. Additionally, visualization on the existing dataset is able to provide better insights and differentiation capabilities.

One aspect I am particularly concerned about is the scalability of this visualization. Currently, the data volume is relatively small, but with a large number of categories, how can the scalability of this visualization be ensured? I would suggest that the authors provide more discussion and experimental results on this point. When the dataset is large, the entire Voronoi becomes complex, with very small cells and densely packed ridges that obscure each other, making it visually unfriendly. I doubt whether users other than authors can derive similar insights from such dense and non-interactive visualizations. The authors could provide quantitative and qualitative user surveys to demonstrate the effectiveness and usability of their method. The authors could consider further optimizations, such as clustering before segmentation or refining the segmentation rules, rather than just using Voronoi tessellation.
The visualization method seems to lack interactivity. Providing an interactive tool would enhance the appeal of this work. Additionally, the authors do not explicitly mention which dimensionality reduction method was used. Different dimensionality reduction techniques may affect subsequent Voronoi and decision boundary construction. The authors should provide explanations and evaluations of this aspect.

**Strengths:**

I’m glad to see that some work at the ICLR conference focuses on data visualization. The insights provided by data visualization can better help users understand some of the underlying aspects behind the models. Additionally, visualization on the existing dataset is able to provide better insights and differentiation capabilities.

**Weaknesses:**

One aspect I am particularly concerned about is the scalability of this visualization. Currently, the data volume is relatively small, but with a large number of categories, how can the scalability of this visualization be ensured? I would suggest that the authors provide more discussion and experimental results on this point.

**Questions:**

One aspect I am particularly concerned about is the scalability of this visualization. Currently, the data volume is relatively small, but with a large number of categories, how can the scalability of this visualization be ensured? I would suggest that the authors provide more discussion and experimental results on this point. When the dataset is large, the entire Voronoi becomes complex, with very small cells and densely packed ridges that obscure each other, making it visually unfriendly. I doubt whether users other than authors can derive similar insights from such dense and non-interactive visualizations. The authors could provide quantitative and qualitative user surveys to demonstrate the effectiveness and usability of their method. The authors could consider further optimizations, such as clustering before segmentation or refining the segmentation rules, rather than just using Voronoi tessellation.
The visualization method seems to lack interactivity. Providing an interactive tool would enhance the appeal of this work. Additionally, the authors do not explicitly mention which dimensionality reduction method was used. Different dimensionality reduction techniques may affect subsequent Voronoi and decision boundary construction. The authors should provide explanations and evaluations of this aspect.

---

> ### Author Response · Authors · 2024-11-26
>
> We sincerely appreciate the reviewer **8SHQ** for the comment.
>
> ### **Q1: Scalability of the visualization method**
>
> We believe that bigger dataset sizes won’t cause a significant problem for our approach. If the dataset is exceedingly large, a subset of it can be used for visualization. On a smaller dataset that is independently and identically distributed (i.i.d.) with the larger dataset, the decision boundary of the trained model shows relatively small differences compared to the one trained on the larger dataset. We computed the prediction differences on the test set and the overlap of decision boundaries (formed by predicted ridges) on the test set between models trained on smaller datasets of varying sizes and the **Visualization Model (VM)** trained on the whole pool dataset. The experimental results are as follows:
>
> #### - Prediction Differences:
>
> | Model trained on                  | KL Divergence↓ | JS Divergence↓ | Wasserstein Distance↓ |
> |--------------------------|----------------|----------------|------------------------|
> | **MNIST (1000 samples)** | 0.1890         | 0.0324         | 0.0067                |
> | **MNIST (5000 samples)** | 0.0515         | 0.0101         | 0.0028                |
> | **MNIST (10000 samples)**| 0.0362         | 0.0067         | 0.0019                |
> | **CIFAR-10 (1000 samples)** | 0.1255      | 0.0318         | 0.0121                |
> | **CIFAR-10 (5000 samples)** | 0.0371      | 0.0094         | 0.0045                |
> | **CIFAR-10 (10000 samples)**| 0.0177      | 0.0043         | 0.0025                |
>
> The smaller the values of KL Divergence, JS Divergence, and Wasserstein Distance, the more similar the two predicted probability distributions are.
>
> #### - Overlapping Predicted Ridges:
>
>
> | Model trained on        | Total Predicted Ridges↓ | Overlapping Predicted Ridges (compared to VM)↑ |
> |--------------------------|----------------|--------------------------------------|
> | **CIFAR-10 (1000 samples)** | 3728         | 2587                                 |
> | **CIFAR-10 (5000 samples)** | 3479         | 3072                                 |
> | **CIFAR-10 (10000 samples)**| 3520         | 3233                                 |
> | **CIFAR-10 (40000 samples) VM** | 3426      | 3426                                 |
>
>
> | Model trained on        | Total Predicted Ridges↓ | Overlapping Predicted Ridges (compared to VM)↑ |
> |--------------------------|----------------|--------------------------------------|
> | **MNIST (1000 samples)** | 2919          | 1224                                 |
> | **MNIST (5000 samples)** | 1936          | 1435                                 |
> | **MNIST (10000 samples)**| 1880          | 1518                                 |
> | **MNIST (50000 samples) VM** | 1799       | 1799                                 |
>
> Additionally, we will fill different cells with colors to make it easier to observe the changes in regions. In fact, in our paper, we constructed 50,000 Voronoi cells and 40,000 Voronoi cells, respectively.
>
> The real issue is likely to arise when there are many classes. Anything beyond approximately 20 classes becomes very challenging to visualize. To address this, we recommend splitting the task into multiple sub-classification tasks and performing visualization on these smaller subsets. Our approach mainly focuses on exploring the sampling behaviors of classical query strategies and analyzing the impact of newly sampled points on the model’s training in the subsequent round.
>
> ### **Q2: Add quantitative and qualitative user surveys**
>
> Conducting qualitative or quantitative user surveys based on our current visualization exploration of different query strategies in active learning is indeed not feasible for us at this stage. In this paper, we primarily performed qualitative analysis of the visualization results we obtained.
>
> However, we greatly appreciate your suggestion regarding interactivity—designing an interactive visualization tool for active learning is indeed part of our future work. When implementing the interactive tool in the next step, we will consider conducting relevant user studies.

---

> ### Author Response · Authors · 2024-11-26
>
> ### **Q3: Effect of different dimensionality reduction methods on Voronoi and decision boundary**
>
> We believe the point you raised is highly valuable, especially in the context of low-dimensional visualizations. Different dimensionality reduction methods result in varying degrees of information loss, which can cause spatial distortion in our visualization results and, consequently, affect the decision boundary constructed in the reduced dimensions. However, this spatial distortion is an inherent and objective issue, and its impact is limited according to the experimental results. Our visualization method does not rely on any specific dimensionality reduction technique, and exploring the effect caused by different dimensionality reduction methods across various datasets is not the core focus of this work. However, in the revision, we will include an analysis of the effect caused by the dimensionality reduction technique currently employed (t-SNE) in our method.
>
> We used two different methods for evaluation:
>
> 1. **Correlation Test** [1, 2]:
>    This method evaluates the Pearson correlation between the pairwise similarity of the features extracted by the model and the pairwise distance matrix calculated from the dimensionality-reduced data. The results are shown in the table below:
>
>    | &#124;Pearson Correlation&#124;↑     | EntropySampling (Avg) | RandomSampling (Avg) | Visualization Model |
>    |--------------------------|-----------------------|-----------------------|---------------------|
>    | **MNIST**               | 0.6307               | 0.6630               | 0.5951             |
>    | **CIFAR-10**            | 0.5049               | 0.4615               | 0.5166             |
>
>    Since sampling occurs over multiple rounds, we calculated the average values. The absolute value of Pearson Correlation closer to 1 indicates a higher correlation, while values closer to 0 indicate no correlation. The results indicate that the impact of spatial distortion caused by dimensionality reduction is relatively small.
>
> 2. **Local Structure Preservation** [3]:
>    We trained and tested an SVM on the same training data as in each round of the AL process, but used the corresponding dimensionality-reduced 2D features for visualization instead of the original data, and compared its accuracy with that of the model in the AL process trained on the original data. Additionally, we evaluated the results of a 1-NN classifier on the 2D test set. The results are as follows:
> #### - Accuracy:
>    | Round | AL RandomSampling (CIFAR-10) | SVM     | 1-NN    |
>    |-------|-------------------------------|---------|---------|
>    | 0     | 0.3399                        | 0.4186  | 0.6467  |
>    | 1     | 0.5191                        | 0.8096  | 0.8817  |
>    | 2     | 0.6722                        | 0.8931  | 0.8768  |
>    | 3     | 0.7401                        | 0.9080  | 0.8897  |
>    | 4     | 0.7797                        | 0.9269  | 0.9044  |
>    | 5     | 0.8167                        | 0.9246  | 0.8985  |
>    | 6     | 0.8235                        | 0.9288  | 0.9182  |
>    | 7     | 0.8434                        | 0.9320  | 0.9259  |
>    | 8     | 0.8682                        | 0.9414  | 0.9253  |
>    | 9     | 0.8814                        | 0.9411  | 0.9283  |
>    | 10    | 0.8893                        | 0.9380  | 0.9388  |
>    | 11    | 0.8895                        | 0.9499  | 0.9379  |
>
>    It can be observed that both SVM and 1-NN achieve better performance compared to the AL process, indicating that the dimensionality reduction method effectively preserves the local structural characteristics of the original data. More experimental results will be included in the appendix.
>
> #### References
>
> [1] Smyth, Barry, Mark Mullins, and Elizabeth McKenna. "Picture Perfect: Visualisation Techniques for Case-based Reasoning." *ECAI*. 2000.
> [2] Namee, Brian Mac, and Sarah Jane Delany. "CBTV: Visualising case bases for similarity measure design and selection." *International conference on case-based reasoning*. Berlin, Heidelberg: Springer Berlin Heidelberg, 2010.
> [3] Huang, Haiyang, et al. "Towards a comprehensive evaluation of dimension reduction methods for transcriptomic data visualization." *Communications Biology* 5.1 (2022): 719.

---

### Official Review · Reviewer_Vpbn · 2024-11-02

**Soundness:** 2
**Presentation:** 2
**Contribution:** 2
**Rating:** 3
**Confidence:** 4

**Summary:**

The proposed work deals with the challenging issue of getting insights into sampling strategies of active learning. Addressing this issue, the authors developed a novel visualization approach: The feature space of the samples is projected to a 2D plane and segmented into Voronoi cells. Ridges of the Voronoi cells are used to construct the decision boundary. Additionally, the confidence of this decision boundary is assessed by leveraging predicted probabilities of samples for each ridge. Thus, the confidence of the decision boundary varies locally. The authors apply the confidence decision boundary to visualize the learning and sampling behavior of eight active learning strategies on MNIST and CIFAR-10. Using various other visualizations, they qualitatively depict the characteristics of the sampling strategies and conclude that uncertainty caused by insufficient training samples may be effectively tackled by sampling. In contrast, uncertainty in noisy regions may be hard to tackle.

**Strengths:**

- The paper sheds light on the crucial topic of getting insights into machine learning algorithms and provides visual results that highlight apparent differences between sampling strategies.
- The chosen approach of this paper is generalizable to various machine learning models performing prediction tasks.
- The authors provide extensive visualizations for each sampling strategy and compare their characteristics with the help of side-by-side visualizations.
- The paper examines a high number of sampling strategies.

**Weaknesses:**

- The paper contains errors in writing. For example, the sentence "Since points on either side of the predicted ridges belong to different predicted classes, the confidence of the predictions vary, different sections of the decision boundary carry varying degrees of informative value due to differences in prediction confidence." in lines 110-113 should be corrected.
- Visualizations lack proper color encoding. For example, the decision boundary, a continuous value, is represented with a qualitative color scale (see Figure 5).
- It remains unclear which snapshots of the feature space were taken for visual elaboration. For example, it is not described why the third round of training was used to inspect the decision boundary in Figure 5. Thus, the authors' conclusions are only based on single visual findings, making them hard to retrace.
- A clear methodology for selecting snapshots for inspection would improve the clarity and usefulness of the authors' approach. Furthermore, the evaluation relies on observations of a single training run - a more extensive evaluation with a higher number of training runs per sampling strategy would be a good starting point.
- Some visualizations lack proper annotations, making them hard to understand. Figure 4 especially needs annotations on what the different colors encode. Similarly, which training rounds do we see in Figure 2?

**Questions:**

- It is unclear why one can assess the confidence of the predicted ridges with the representative Voronoi center points $\mathbf{p_i}$ and $\mathbf{p_j}$ (equation 2). Although each ridge can be represented by its representative point $\mathbf{p}$, the predicted probabilities close to the decision boundary may differ strongly from their center. Could you please clarify why using the point $\mathbf{p}$ gives an accurate prediction of the confidence?
- How did you decide on which snapshots to visualize?

---

> ### Author Response · Authors · 2024-11-26
>
> We sincerely appreciate the reviewer **Vpbn** for the comment.
>
> ### **Q1: Writing errors**
>
> Thank you for pointing out the issues. We will rewrite these relatively complex sentences to achieve better clarity and expression.
>
>
> ### **Q2: Visualizations lack proper color encoding**
>
> It is true that the confidence of the decision boundary (represented by all predicted ridges in our visualization results) is a continuous value. However, because there are too many predicted ridges, using continuous values for color mapping would result in similar colors, making it difficult to distinguish high-confidence and low-confidence regions.
>
> Using intervals to assign colors creates a stronger contrast effect, which enhances observation clarity.
>
>
> ### **Q3: Snapshots selection and the need for a clear methodology for selecting snapshots for inspection**
>
> We did not visualize snapshots of specific rounds but instead visualized the entire active learning process. The examples shown in the main text are only a part of the results. We will organize these visualizations and include them in the appendix of the revised version.
>
> Figure 5 selects the third round because the second row of Figure 5 shows the cumulative sampling points from rounds 1 to 5. Therefore, we chose the decision boundary of the middle round (round 3) for display. As a result, the third row shows the error detection of round 4 to observe the impact of the points selected by the model trained in round 3 on the training of the model in the next round.
>
>
> ### **Q4: Why one can assess the confidence of the predicted ridges with the representative Voronoi center points Pi and Pj (equation 2). Although each ridge can be represented by its representative point P, the predicted probabilities close to the decision boundary may differ strongly from their center.**
>
> You may have misunderstood the Voronoi point \( P \), the Voronoi cell, and the ridge. According to the properties of Voronoi tessellation, the Voronoi point \( P \) can represent all points that fall within its associated Voronoi cell (see lines 119-120 of the paper). Moreover, \( P_i \) and \( P_j \) are the two closest points from our existing point set to the ridge.
>
> The situation you mentioned, where the point closest to the predicted ridge may be very different from the representative point of its Voronoi cell, can indeed occur. However, the Voronoi cell can be regarded as a collection of similar points, and the representative Voronoi point can represent all the points falling within its associated cell based on the nearest neighbor relationship (for details, see Chapter 7 of *Computational Geometry: Algorithms and Applications*) [1].
>
> In our experiments, the Voronoi points correspond to the points already learned by the AL model, while the points within the cells are those from the pool dataset that the model has not yet learned. Therefore, this situation does not affect our evaluation of the confidence of the ridge dividing two different-class cells.
>
> #### Reference
> [1] De Berg, Mark. *Computational geometry: algorithms and applications*. Springer Science & Business Media, 2000.

---

> > ### Comment · Reviewer_Vpbn · 2024-11-30
> >
> > I thank the authors for clarifying some of my concerns and improving the manuscript. However, I still disagree with the choice of color encodings for some visualizations:
> >
> > - Using a qualitative color scale for a continuous value neglects the similarity between values.
> > - Introducing intervals for a continuous value produces potential biases as the interval boundaries might change the entire color encoding. The interval boundaries are chosen arbitrarily.
> >
> > I keep to the score of reject.

---

> > > ### Author Response · Authors · 2024-11-30
> > >
> > > We appreciate the reviewer **Vpbn's** feedback and understand the concerns regarding the potential arbitrariness of interval boundaries. However, we chose to use interval-based color encoding after carefully considering and testing different approaches, including a continuous color scale. Our findings are as follows:
> > >
> > > 1. **Challenges with Continuous Color Encoding:**
> > >    While continuous color encoding can theoretically reflect the similarity between values, in our specific context, it presented challenges. Across multiple rounds, the confidence levels of ridges showed only marginal overall improvements, resulting in subtle variations that were difficult to distinguish when using a continuous color scale. This approach made it harder to observe the between-rounds changes in decision boundary confidence clearly, as the lack of contrast reduced the visualization's ability to highlight key differences between rounds.
> > >
> > > 2. **Advantages of Interval-Based Encoding:**
> > >    To address the above challenge, we introduced interval-based encoding with a 0.1 step size, enhancing the visualization's clarity and making it easier to compare decision boundary changes between rounds. Through extensive testing, we found that finer intervals (e.g., 0.05) introduced excessive granularity, complicating the analysis. Conversely, coarser intervals (e.g., 0.2) oversimplified the data, reducing the effectiveness of the visualization. A 0.1 interval struck the optimal balance.
> > >
> > > While any interval-based approach may introduce some degree of boundary sensitivity, our choice of 0.1 intervals was not arbitrary. It was informed by repeated experimentation and evaluation to ensure it supported effective visualization and interpretation of the decision boundary's confidence levels.

---

### Official Review · Reviewer_kBv3 · 2024-11-02

**Soundness:** 2
**Presentation:** 2
**Contribution:** 3
**Rating:** 6
**Confidence:** 3

**Summary:**

This paper introduces a Voronoi tessellation-based visualization method for active learning (AL) to address the limitations of traditional visualizations, which often lack detail in uncertain or sparse boundary areas. By using a ridge confidence metric with Voronoi cells, the approach offers a clearer, more informative view of decision boundaries, aiding in analyzing different AL sampling strategies. Tested on MNIST and CIFAR-10 datasets, the method reveals unique sampling behaviors across entropy, margin, and diversity-based strategies, helping clarify how each handles uncertainty and impacts model learning.

**Strengths:**

1. By using Voronoi tessellation and introducing a ridge confidence metric, the paper provides a more detailed and interpretable visualization of decision boundaries, offering insights into the complexity and uncertainty of boundary regions.
1. The authors compare multiple AL sampling methods (e.g., entropy, margin, BALD dropout, KMeans) on distinct datasets, providing valuable insights into the behaviors and trade-offs of each approach in different scenarios.
1. The visualization effectively demonstrates how models handle uncertainty due to limited training samples and noisy regions, which is beneficial for identifying optimal query strategies for different types of uncertainties.
1. While focused on active learning, the proposed visualization technique has potential utility in other areas of machine learning that require understanding complex decision boundaries in high-dimensional spaces.

**Weaknesses:**

1. This paper lacks a review of related work. It's important to examine the literature in the related fields, especially other visualization methods developed for active learning.
1. While the work in the paper is valuable, it also lacks of the baseline methods. The paper demonstrates the effectiveness of Voronoi tessellation and ridge confidence in active learning throught several case studies, but it does not prove how much better it is compared with other visualization methods. I suggest a quantitative analysis with different visualization methods, like a user study.
1. Section 3 is over lengthy. The authors should consider breaking it down. Subsection 3.1 and Subsection 3.2 should be independent sections.
1. The discussion of the 2D feature map is missing. How much does space distortion affect the Voronoi tesselletion construction? The neighbors in the original space are not always the same in the feature space. How reliable is the decision boundary in this case? Since everything is visualized in the feature map, it's essential to give a comprehensive discussion on the feature map itself.
1. The visualization strategies are informative, but the implementation in the plots are really hard to follow. I have several suggestions for improvement:
    - Figure 2, 3, 5, 6, 7, and 8 are visualization results, but they are too small and dense to read, which is fatal for a visualization paper. At least the authors should put high resolution images in the appendix.
    - In figure 5, the colormap of confidence interval should be different from that of the classes. I suggest different levels of gray to show the confidence interval
    - In the visualization plots, the scatter points, Voronoi ridges, and cells are crowded and overlapping, causing severe visual clutter. Actually, it's not necessary to show all the information in a single plot at once. It only makes the plot massive. One way to display a large quantity of information is to enable user interactions. For example, enable users to choose classes they are interested in, add a slider to show different levels of confidence, brush to zoom in on a decision boundary.
    - The Error Detection visualization results are interesting. It gives more information on how the active learning model behaves. I suggest putting multiple training iterations of Error Detection plots in the appendix to visualize its evolution over time.

While I believe this work makes good contribution, I'm afraid the authors cannot resolve my concerns in a reasonable time. Therefore, I recommend a weak reject.

**Questions:**

I don't quite understand how the 2D feature map is constructed. It is said that it directly comes from the neural network model, but how? Could the authors provide more details?

---

> ### Author Response · Authors · 2024-11-26
>
> We sincerely appreciate the reviewer **kBv3** for the comment.
>
> ### **Q1: Lack of related work and over length of section 3**
>
> There are very few visualization methods specifically targeting **active learning (AL)**, and most are limited to basic visualizations accompanying the proposal of query strategies. Generalizable decision boundary visualization methods are also scarce.
>
> Due to the limited page length required by ICLR and based on the format of similar ICLR papers in previous years (for instance, Jenssen, Robert. "MAP IT to Visualize Representations." *The Twelfth International Conference on Learning Representations*), we integrated the related work section into the introduction section.
>
> Regarding the issue you mentioned about Section 3 being too long, we will restructure the paper in the revised version, splitting the original Subsections 3.1 and 3.2 into two independent sections.
>
>
> ### **Q2: Lack of baselines for visualization methods**
>
> Quantitative comparison is indeed a very good suggestion. However, the focus of this paper is to explore the sampling behaviors and results of different query strategies, leading to conclusions that cannot be derived through other visualization methods or direct comparisons. Different visualization methods emphasize different aspects. In the introduction section, we discussed various existing visualization methods and their applicability to active learning.
>
>
> ### **Q3: Construction of the 2D feature map**
>
> In lines 208-232 of the paper, we describe the features used to generate the Voronoi diagram. In fact, we have two ways of constructing a 2D feature map:
>
> 1. **Dynamically Updated Features**: These features are updated during active learning (e.g., Figure 2). They are extracted by the trained model at each round on the pool data and test data, with the dimensionality-reduced results serving as Voronoi points for constructing our visualization.
>
> 2. **Fixed Features for Posterior Analysis**: These features are used for posterior analysis (e.g., Figures 5, 6, and 7). Because we need to observe and compare the sampling behaviors and results of different query strategies, we focus on analyzing the confidence decision boundary, sampling points (newly queried points), and the impact of new samples on model training (error detection) using a fixed feature map. This is necessary due to the differences introduced by the features extracted by models from different rounds and further variations caused by different dimensionality reduction methods.
>
> Thus, the second method involves training a **visualization model** (described in Section 3.1.3) using all the pool data to help generate fixed features, which are then used for the feature map in every round. The reason for using the visualization model is explained in lines 211-215 of the paper.

---

> ### Author Response · Authors · 2024-11-26
>
> ### **Q4: Effects of spatial distortion on Voronoi tessellation construction**
>
> We believe the point you raised is highly valuable, and a comprehensive discussion about the feature map itself is indeed constructive. The impact of spatial distortion is an inherent and objective issue, though its effects are limited according to the experimental results. Our visualization method does not rely on any specific dimensionality reduction technique, and exploring the spatial distortion introduced by different dimensionality reduction methods across various datasets is not the core focus of this work. However, in the revision, we will include an analysis of the spatial distortion caused by the dimensionality reduction technique currently employed (t-SNE) in our method.
>
> We used two different methods for evaluation:
>
> 1. **Correlation Test** [1, 2]:
>    This method evaluates the Pearson correlation between the pairwise similarity of the features extracted by the model and the pairwise distance matrix calculated from the dimensionality-reduced data. The results are shown in the table below:
>
>    | &#124;Pearson Correlation&#124;↑     | EntropySampling (Avg) | RandomSampling (Avg) | Visualization Model |
>    |--------------------------|-----------------------|-----------------------|---------------------|
>    | **MNIST**               | 0.6307               | 0.6630               | 0.5951             |
>    | **CIFAR-10**            | 0.5049               | 0.4615               | 0.5166             |
>
>    Since sampling occurs over multiple rounds, we calculated the average values. The absolute value of Pearson Correlation closer to 1 indicates a higher correlation, while values closer to 0 indicate no correlation. The results indicate that the impact of spatial distortion caused by dimensionality reduction is relatively small.
>
> 2. **Local Structure Preservation** [3]:
>    We trained and tested an SVM on the same training data as in each round of the AL process, but used the corresponding dimensionality-reduced 2D features for visualization instead of the original data, and compared its accuracy with that of the model in the AL process trained on the original data. Additionally, we evaluated the results of a 1-NN classifier on the 2D test set. The results are as follows:
> #### - Accuracy:
>    | Round | AL RandomSampling (CIFAR-10) | SVM     | 1-NN    |
>    |-------|-------------------------------|---------|---------|
>    | 0     | 0.3399                        | 0.4186  | 0.6467  |
>    | 1     | 0.5191                        | 0.8096  | 0.8817  |
>    | 2     | 0.6722                        | 0.8931  | 0.8768  |
>    | 3     | 0.7401                        | 0.9080  | 0.8897  |
>    | 4     | 0.7797                        | 0.9269  | 0.9044  |
>    | 5     | 0.8167                        | 0.9246  | 0.8985  |
>    | 6     | 0.8235                        | 0.9288  | 0.9182  |
>    | 7     | 0.8434                        | 0.9320  | 0.9259  |
>    | 8     | 0.8682                        | 0.9414  | 0.9253  |
>    | 9     | 0.8814                        | 0.9411  | 0.9283  |
>    | 10    | 0.8893                        | 0.9380  | 0.9388  |
>    | 11    | 0.8895                        | 0.9499  | 0.9379  |
>
>    It can be observed that both SVM and 1-NN achieve better performance compared to the AL process, indicating that the dimensionality reduction method effectively preserves the local structural characteristics of the original data. More experimental results will be included in the appendix.
>
> #### References
>
> [1] Smyth, Barry, Mark Mullins, and Elizabeth McKenna. "Picture Perfect: Visualisation Techniques for Case-based Reasoning." *ECAI*. 2000.
> [2] Namee, Brian Mac, and Sarah Jane Delany. "CBTV: Visualising case bases for similarity measure design and selection." *International conference on case-based reasoning*. Berlin, Heidelberg: Springer Berlin Heidelberg, 2010.
> [3] Huang, Haiyang, et al. "Towards a comprehensive evaluation of dimension reduction methods for transcriptomic data visualization." *Communications Biology* 5.1 (2022): 719.
>
> ### **Q5: Presentation of visualization results**
>
> We sincerely thank you once again for your suggestions regarding the presentation of visualization results. We will include more high-resolution figures in the appendix and revise the way figures are presented in the main text.
>
> This paper primarily aims to leverage visualization to explore the sampling behavior characteristics of some classical query strategies. Based on this, developing a visualization tool that enables interactive and dynamic observation of AL is our next step.

---

> > ### Comment · Reviewer_kBv3 · 2024-11-29
> >
> > I appreciate the authors’ response and their thorough analysis of spatial distortion, which has effectively addressed my concerns. However, as a visualization paper, it is crucial to ensure that the visualization scheme is efficient. I strongly recommend that the authors revisit their design goals and refine the visualization tools to enhance their usability and clarity. Based on this, I would raise the grade from 5 to 6.

---

### Meta-Review · Area_Chair_Wqf6 · 2024-12-20

**Metareview:**

**(a) Summary**

This paper presents a novel visualization approach for active learning using Voronoi tessellation and a ridge confidence metric. The method projects high-dimensional feature spaces onto 2D planes and constructs decision boundaries using Voronoi cell ridges. The authors apply this visualization technique to analyze eight different active learning sampling strategies on MNIST and CIFAR-10 datasets, revealing distinct sampling behaviors and their effectiveness in handling different types of uncertainties.

**(b) Strengths**
- Provides detailed and interpretable visualization of decision boundaries through Voronoi tessellation
Effectively demonstrates model behavior in handling different types of uncertainties
- Successfully provides insights into ML models through comprehensive comparison of sampling strategies

**(c) Weaknesses**
- Limited literature review and baseline comparisons
- Lack of user studies to validate visualization effectiveness
- Scalability challenges with larger datasets
- Suboptimal visualization design choices, particularly in color encoding

**(d) Decision factors**

While the paper presents an interesting visualization approach for active learning, several key issues remain unresolved. The visualization design choices need refinement, particularly in color encoding and scalability. The lack of user studies and baseline comparisons makes it difficult to validate the method's practical utility. While we acknowledge the potential value of this approach, these limitations lead to a recommendation for rejection.

**Additional Comments On Reviewer Discussion:**

Although the authors provided detailed responses regarding spatial distortion analysis, two reviewers (kBv3 and Vpbn) remained unconvinced about the fundamental visualization design choices, particularly the color encoding schemes and representation of continuous values. While two reviewers (kBv3 and rwsQ) acknowledged the improvements and raised their score, the consensus was that the visualization design and experimental evaluation require significant refinement, especially in terms of usability and clarity, before it can effectively serve as a visualization tool.

---

### Decision · Program_Chairs · 2025-01-22

Reject